# Highly efficient expression of DNA-peptide conjugates in growth-arrested cells

Zulfiqar Y. Mohamedshah[1], Chih-Chin Chi[1], Ember M. Tota[1], Alexis C. Komor [1,2,3,4] & Neal K. Devaraj [1,2,5] ✉

Efficient nuclear delivery of DNA remains a major challenge in non-viral gene therapy. While nuclear localization signal (NLS) peptides have been explored for enhancing nuclear translocation of DNA, their efficacy has been limited by DNA-peptide conjugation strategies. Leveraging *E. coli* tRNA guanine transglycosylase, we present a modular workflow for generating DNA oligonucleotide-peptide conjugates which are ligated to linear DNA to generate peptide-modified gene cassettes (DNA-PepTAG). Using an eGFP reporter delivered via lipofection to growth-arrested cells, NLS-modified gene cassettes significantly increases nuclear localization, mRNA transcription, and expression up to ~10 fold compared to unmodified gene cassettes. Screening multiple NLS peptides in growth-arrested human cell lines reveal cell-type-specific preferences for nuclear translocation of DNA cargo. Two NLS peptides, PLSCR-1 and extSV40, exhibit consistently high expression across tested cell types, indicating broad applicability for nuclear delivery. We evaluate the generality of our approach by delivering DNA payloads encoding for both cytosolic and secreted proteins, as well as gene cassettes ranging in size from 1.3 kbp to 7 kbp. These findings support the potential of DNA-NLS conjugates as a viable strategy for non-viral gene therapy, enabling enhanced nuclear delivery of therapeutic genes while minimizing the required DNA dose.

The effective delivery of exogenous DNA into cells is necessary for modern medical and technical advances including gene therapy, antisense oligonucleotides, and many genome editing applications such as CRISPR-Cas9 and related systems[1–4]. While viral methodologies for therapeutic DNA delivery have shown promise, significant safety concerns remain regarding immunogenicity, insertional mutagenesis, toxicity, and off target effects[5–9]. Efficacy has also been limited due to restricted cargo capacity, scalability issues, and low targeting specificity[5–8,10]. The success of mRNA based COVID-19 vaccines delivered through lipid nanoparticles (LNPs) has motivated the application of non-viral delivery methods for DNA therapeutics[11–15]. However, unlike mRNA which requires only entry into the cytosol, DNA therapeutics must be delivered to the nucleus. DNA can offer a more stable therapeutic platform than RNA, having greater chemical stability and potentially longer-lasting expression[16]. Non-viral techniques, including transfection of naked DNA, remain highly inefficient in this regard, and it is believed that only ~1% of DNA transfected in mammalian cells is translocated to the nucleus[17–19]. A majority of DNA enters the nucleus during cell division through the breakdown and subsequent reformation of the nuclear envelope[17–20]. Thus, DNA gene therapy typically requires very high doses of DNA which has hindered biological and medical applications due to cytotoxicity and the potential to trigger an immune response[7,17–19,21–23].

Conjugation of DNA to nuclear localization signals (NLSs) has been explored for improving nuclear delivery of DNA cargo[24–32]. NLSs

[1]Department of Chemistry and Biochemistry, University of California San Diego, La Jolla, CA, USA. [2]Department of Biochemistry and Molecular Biophysics, University of California San Diego, La Jolla, CA, USA. [3]Moores UCSD Cancer Center, University of California San Diego, La Jolla, CA, USA. [4]Sanford Stem Cell Institute, University of California San Diego, La Jolla, CA, USA. [5]Department of Bioengineering, University of California San Diego, La Jolla, CA, USA. ✉e-mail: ndevaraj@ucsd.edu

are small peptide fragments derived from eukaryotic nuclear or viral proteins that facilitate the import of a protein cargo into the nucleus through nuclear pore complexes[33–35]. Classical NLSs are defined by a short amino acid sequence with one or more clusters of basic residues that are recognized by importin-α which is subsequently recognized by importin-β, facilitating translocation of the importin-protein-NLS complex to the nucleus[34]. While previous attempts to generate DNA-NLS conjugates for nuclear delivery demonstrated increased nuclear translocation and subsequent expression, results have been inconsistent and often difficult to reproduce[24–32]. One possible reason for these inconsistences is the variation in the conjugation method attaching the NLS to the DNA cargo, which likely impacts nuclear translocation (Fig. 1a). Electrostatic interactions between NLS peptides (or proteins containing NLS peptides) and DNA may hinder cargo translocation by limiting NLS accessibility to its transport partners[24,28,32,36,37]. Furthermore, current non-covalent strategies to form DNA-NLS conjugates have the potential to dissociate in biological contexts and their covalent counterparts typically rely on labile linkages or non-site-specific modification that can lead to multiple NLS peptides per DNA cargo[24,27–29,31,35,37]. Other factors, including location of the NLS on the DNA, the identity and size of the DNA cargo, linker length between the NLS and the DNA cargo, and type and number of NLS peptides likely also have an impact on nuclear translocation of DNA[24,28]. A site-specific, irreversible covalent conjugation strategy between DNA and peptides that can be rapidly implemented is required to determine if DNA-NLS conjugation is a reliable strategy for increased nuclear translocation of a DNA cargo.

Most strategies for generating oligonucleotide peptide conjugates are laborious, involve complex and harsh reaction/purification conditions, and suffer from poor yields[38–41]. This is likely why previous studies have explored only a limited repertoire of DNA-NLS species[24–31]. Furthermore, the frequently utilized strategy of thiol-maleimide conjugation (Fig. 1a) suffers from reaction reversibility[42]. To circumvent these issues, we developed a high-yielding chemoenzymatic workflow to generate DNA-peptide conjugates that leverages the activity of bacterial (*E. coli*) tRNA guanine transglycosylase (TGT) to modify DNA with functional probes[43–45]. We have previously shown that TGT is able to exchange a specific guanine base in DNA/RNA stem loops with chemically modified preQ$_1$ probes, referred to as DNA /RNA-TAG[43–45]. Here we introduce DNA-PepTAG, where we expand DNA-TAG technology to generate peptide-modified gene cassettes. TGT is employed to site-specifically insert preQ1-strained-cyclooctynes into DNA oligonucleotides, generating bioconjugation substrates for strain-promoted azide-alkyne cycloaddition (SPAAC) to azide-modified peptides and creating a modular workflow for DNA oligonucleotide-peptide conjugation. These DNA oligonucleotide-peptide conjugates are then ligated to the ends of linear DNA, creating a gene cassette site-specifically modified with a peptide.

Utilizing DNA-PepTAG, we have developed a workflow in which an eGFP reporter gene cassette is conjugated with NLS peptide(s), allowing for facile screening of different NLS sequences, conjugation architectures, and linkers (Fig. 1b). As exogenous DNA is thought to enter the nucleus primarily during cell division, we transfect, via Lipofectamine 3000, NLS-modified eGFP gene cassettes into growth-arrested human cell lines to isolate nuclear translocation mediated solely by NLS activity[17–20]. Unlike some previous studies that have used actively dividing mammalian cells, the use of growth-arrested cells allows direct assessment of the efficacy of NLS peptides in facilitating nuclear entry. Monitoring eGFP expression, we identify an optimal amino acid (GGGGS)$_3$ flexible linker length between the DNA cargo and

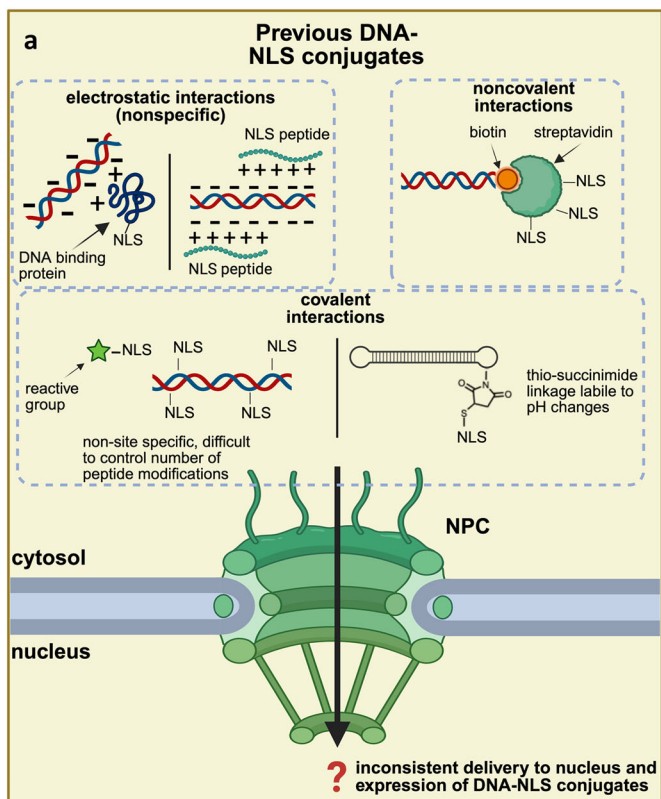
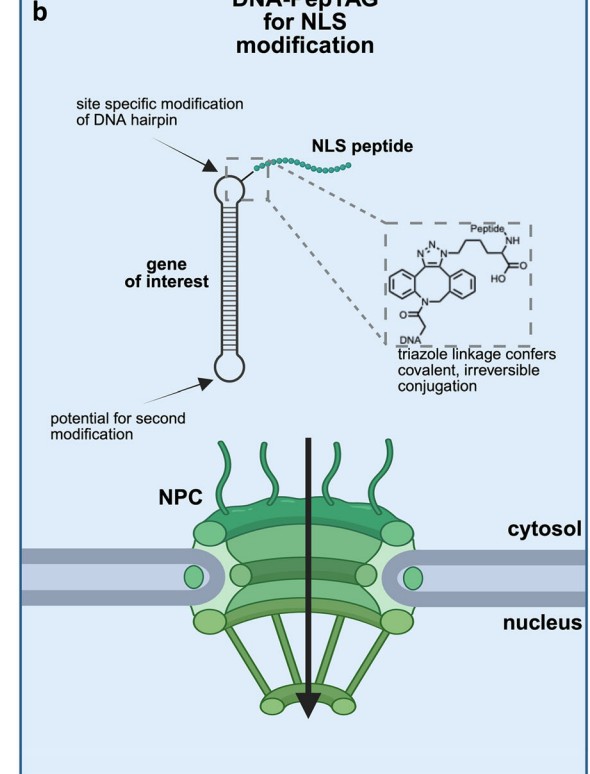

**Fig. 1 | An overview of DNA-NLS conjugates for nuclear translocation of DNA cargos. a** Summary of DNA-NLS conjugate strategies previously explored for nuclear translocation of a DNA cargo through the nuclear pore complex (NPC). DNA-NLS conjugates are summarized in three broad categories: NLS-tagged nuclear proteins associated electrostatically with DNA cargo, noncovalent interactions between DNA cargo and NLS peptides, and covalent interactions between DNA cargo and NLS peptides. **b** Overview of NLS-modified gene cassettes generated by DNA-PepTAG. DNA-PepTAG conjugates rely on a nonreversible, biocompatible triazole linkage between the NLS peptide and DNA cargo of interest. Created in BioRender. Mohamedshah, Z. (https://BioRender.com/xkf71lx).

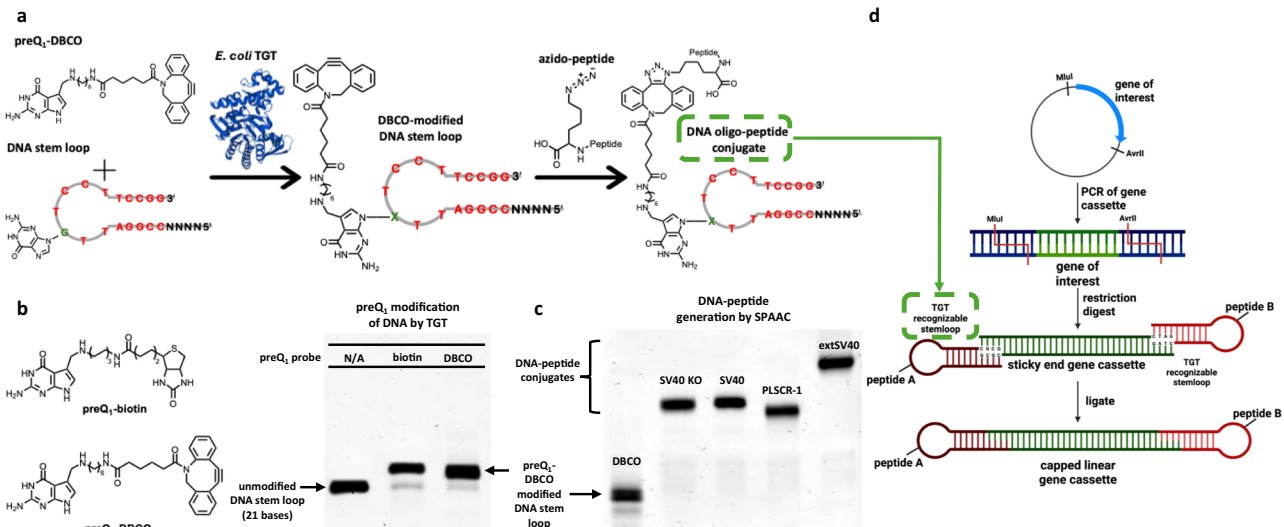

**Fig. 2 | Peptide conjugation to DNA using DNA-PepTAG. a** General workflow to generate DNA oligonucleotide-peptide conjugates. *E.coli* TGT irreversibly exchanges a specific guanine with a preQ₁-DBCO in a DNA oligonucleotide stem loop to generate a DBCO-modified DNA stem loop. Strain-promoted azide-alkyne cycloaddition is then used to form a triazole linkage between the DBCO-modified DNA stem loop and an azido-peptide, generating a DNA oligonucleotide-peptide conjugate. The minimal TGT stem loop is highlighted in red with the target guanine in green. The stem loop can be extended at both the 5′ and 3′ ends; in this work the 5′ end was extended to match the required restriction enzyme sticky end (i.e., MluI). **b** Chemical structures of preQ₁ probes (left) and their covalent attachment to a DNA stem loop via TGT by Urea-PAGE (right). PreQ₁-biotin serves as a positive control for DNA-TAG activity. The upward gel shift between the negative control (MluI 170 PP DNA hairpin, lane 1) and preQ₁-biotin (lane 2) and preQ₁-DBCO (lane 3) suggests near quantitative yields. **c** Urea-PAGE analysis of strain-promoted azide-alkyne cycloaddition reactions between DBCO-modified DNA stem loop and

various azido-peptide NLSs. The upward gel shifts indicate formation of the DNA oligo-peptide conjugates with near quantitative yields. Lane 1 – DBCO modified MluI 170 PP DNA hairpin, Lane 2 – SV40 KO Lx3, Lane 3 – SV40 NLS LX3, Lane 4 – PLSCR-1 NLS Lx3, Lane 5 – extSV40 NLS Lx3. Gels in (b) and (c) are representatives of 3 independent experiments. Source Data are provided as a Source Data file. **d** General scheme to generate peptide modified linear gene cassettes for mammalian cell expression. The gene of interest (including gene, enhancer, promoter, poly-A tail, and flanking restriction sites) is PCR amplified from plasmid DNA. The PCR product is then dual digested with relevant restriction enzymes (MluI and AvrII) to generate defined sticky ends on the PCR products. DNA stem loops designed with complimentary sticky ends may be modified with peptides (peptide A and/or B) and ligated to the sticky end gene fragment to generate a capped linear gene cassette. Created in BioRender. Mohamedshah, Z. (https://BioRender.com/nfcakg2).

NLS peptide. Interestingly, we find that appending the NLS to a single side of the cassette significantly enhances expression, whereas dual modifications on opposite ends drastically reduces expression to near baseline levels. We quantify the copy number of exogenous DNA translocated to the nucleus and find that NLS-modified gene cassettes had significantly enhanced nuclear delivery compared to controls. Screening a panel of NLS peptides across three growth-arrested cell lines reveals that, while all peptides enhance eGFP expression compared to the unmodified gene cassette, their efficacy varies by cell type, suggesting cell/tissue specificity of NLS peptides for nuclear delivery of DNA cargo. Finally, NLS-modified *Factor IX* gene cassettes exhibit ~10-fold higher secretion than controls, highlighting the potential of DNA-NLS conjugates for non-viral gene therapy applications.

## Results and Discussion
### Peptide conjugation to DNA using DNA-PepTAG
The developed workflow to generate DNA oligonucleotide-peptide conjugates using DNA-TAG is shown in Fig. 2a. The modular nature of the platform allows functionalization of a custom DNA oligonucleotide sequence with preQ₁-DBCO, a strained cyclooctyne commonly used for click chemistry, as long as the minimal TGT recognition stem loop is present (Fig. 2a). Once covalently modified with a DBCO moiety, azide-containing peptides are "clicked" to the functionalized DNA oligonucleotide. DNA oligonucleotide-peptide conjugates can then be ligated to linear DNA to generate peptide-modified gene cassettes.

A preQ₁-DBCO probe was synthesized, purified, and characterized as previously described for other preQ₁ derivatives[43,44]. The preQ₁-DBCO probe was incubated with TGT and the MluI 170 PP DNA oligonucleotide stem loop substrate listed in the Supplementary

Sequences. This stem loop substrate consists of the minimal TGT recognition stem loop and an additional 4-nt sequence on the 5′ end complementary to the sticky end produced by MluI digestion (for ligation to linear DNA in downstream steps). The reaction was analyzed for TGT-mediated DNA modification via denaturing polyacrylamide gel electrophoresis (Urea-PAGE). We observed an upward gel shift of the product relative to the starting material as the mass of the DNA stem loop increased due to the exchange of guanine for preQ₁-DBCO (Fig. 2b, Supplementary Fig. 1, 2). We have previously demonstrated near quantitative yields from modification of DNA stem loops with preQ₁-biotin by TGT, which served as a positive control for TGT activity[43]. TGT-mediated modification of the DNA stem loop with preQ₁-DBCO proceeded with ~90% efficiency, matching that of preQ₁-biotin modification. The use of DBCO allows for simplified reactions conditions compared to unstrained alkynes, particularly the lack of a Cu(I) catalyst required for alkyne-azide click chemistry. Leveraging strain-promoted azide-alkyne cycloaddition (SPAAC) between DBCO-modified DNA and synthesized azido-peptides allows for an orthogonal, non-reversible click reaction in water to form a triazole linkage[46,47]. Following DBCO modification of the DNA stem loop by TGT, addition of azido-peptides to the reaction mixture resulted in the generation of DNA oligonucleotide-peptide conjugates with near quantitative yields (90-95%) after overnight incubation at 37 °C (Fig. 2d, Supplementary Fig. 1, 2). Due to the high-yield generation of DNA oligonucleotide-peptide conjugates, purification was performed via standard DNA oligonucleotide cleaning columns (Zymo Research). The high efficiency of DNA-TAG allows for a low-cost methodology to generate DBCO modified DNA at near quantitative yields[48].

Traditional methods for incorporating click handles into DNA often require the introduction of unnatural nucleobases during solid-

phase phosphoramidite synthesis, leading to increased costs, extensive purification steps, and, in some cases, additional chemical modifications[48–52]. In contrast, our approach uniquely enables site-specific, internal labeling of pre-synthesized, commercially available DNA oligonucleotides with a strained cyclooctyne handle. This methodology, which is not commercially available and remains largely unexplored, streamlines the generation of internally modified, DBCO-labeled DNA oligonucleotides. By leveraging a post-synthetic conjugation strategy, our approach provides a modular and efficient platform to generate DNA oligonucleotide-peptide conjugates via SPAAC[39].

To generate peptide-modified gene cassettes, we chose to utilize a capped linear gene cassette, similar to previous work, that can be selectively modified with peptides to evaluate nuclear uptake of our DNA-NLS constructs (Fig. 2d)[27]. An eGFP gene cassette flanked by two unique restriction sites, AvrII and MluI, was generated via PCR amplification and subsequent digestion, yielding a linear eGFP fragment with defined sticky ends by the restriction enzymes used. Our TGT-recognizable DNA oligonucleotide stem loops were designed such that they contain 5′ overhangs complementary to these sticky ends (either MluI or AvrII), enabling ligation to the digested eGFP fragment to generate a capped eGFP gene cassette which was verified by agarose gel electrophoresis (Supplementary Fig. 3). Using the workflow described, we can modify either or both DNA oligonucleotide stem loops (A and/or B) with azido-peptides prior to ligation, effectively generating single- or dual peptide-modified eGFP gene cassettes following ligation. Negative control samples (capped linear gene cassettes containing no peptide) were generated by ligating unmodified DNA stem loops to both ends of the sticky-ended linear DNA fragment. By leveraging DNA-PepTAG, we rapidly generated NLS-modified gene cassettes, providing a versatile platform for studying peptide-mediated intracellular delivery of DNA cargo.

## Optimization of and nuclear translocation NLS modified gene cassettes for gene expression in growth-arrested cells

We transfected, via Lipofectamine 3000, NLS-modified gene cassettes into aphidicolin growth-arrested HepG2 cells to determine if they would exhibit increased expression (Fig. 3a, b, Supplementary Fig. 3). HepG2 cells are a well-characterized human hepatocellular carcinoma cell line widely used in drug delivery and gene therapy studies due to their metabolic activity and relevance to liver-targeted therapeutics[53,54]. The liver's ability to efficiently take up circulating nucleic acids and drive systemic protein expression has made it a key target for nucleic acid-based therapies, including mRNA vaccines and gene therapies[53–56]. It is understood that a majority of transfected DNA enters the nucleus during the breakdown and subsequent reformation of the nuclear envelope during cell division. We therefore reasoned we could better assess the impact of NLS modifications on DNA nuclear translocation and subsequent expression by growth-arresting HepG2 cells[17–20]. We observed higher eGFP expression from DNA constructs modified with NLS peptides (Fig. 3b) compared to the linear control, which consisted of the same gene cassette ligated to DNA stem loops lacking peptide modification. Notably, the presence of a peptide alone was not sufficient to cause an increase in eGFP expression, as when the "SV40 KO" peptide was used (which differs from the SV40 NLS by a single K→T mutation that has been shown to eliminate nuclear translocation), we observed no increase in eGFP fluorescence compared to the linear control (Fig. 3b)[33].

While we were encouraged by these initial results, we sought to optimize the eGFP expression cassette before screening a number of NLS peptides for their ability to increase nuclear translocation and expression of DNA cargo. In particular, we wanted to determine how the peptide linker length between the NLS and DNA cargo, as well as the number and location of NLS modifications, influences the efficiency of DNA nuclear translocation (Fig. 3a). We used the fraction of

cells with eGFP fluorescence above untransfected controls, quantified by flow cytometry, as a proxy for relative nuclear translocation of DNA. Previous studies have suggested that the distance between the NLS and the DNA has an impact on the nuclear translocation of NLS modified DNA cargos[28,32]. To address this, we transfected 5′ SV40 NLS-modified eGFP expression cassettes with varying lengths of the flexible $(GGGGS)_n$ linker sequences between the NLS portion of the peptide and the azido-lysine into growth-arrested HepG2 cells and monitored expression (Table 1, Supplementary Fig. 4, Supplementary Table 2). Increasing the linker length from $n = 1$ (1x) to $n = 3$ (3x) significantly enhanced eGFP expression (~2-fold increase, $p \leq 0.001$), suggesting that greater peptide-DNA spacing and perhaps charge separation improve NLS-DNA nuclear translocation (Supplementary Fig. 4). However, when the linker length was increased further to $n = 4$ (4x) and $n = 5$ (5x), no additional increase ($p > 0.05$) in eGFP expression was observed, indicating that the 3x linker length is optimal. Furthermore, we examined linker length with a less positively charged NLS (derived from influenza) and observed moderate increases (~1.2-fold) in eGFP fluorescence between the 1x and 2x linker, with no significant increase between the 2x and 3x linker (Supplementary Fig. 4). These results suggest that increasing the flexible linker length, up to an optimal point, enhances DNA-NLS nuclear translocation by improving the accessibility of the NLS peptide to nuclear transport factors, with the increased physical distance, flexibility, and potential charge separation between the DNA and peptide likely contributing to this effect[28,32,36,37].

Unlike previous strategies, DNA-PepTAG enables straightforward and precise NLS modification of DNA gene cassettes at either the 5′ end, 3′ end, or both (Figs. 2d, 3a). We tested eGFP cassettes modified either via the 5′ MluI or, 3′ AvrII sticky ends or, alternatively, modified at both sites with an SV40 or Influenza derived NLS peptide (Fig. 3c). While both single-end modifications had similar ($p > 0.05$) eGFP expression levels to each other (~16% for SV40 NLS, ~18% for Influenza NLS), the dual-end modification drastically reduced expression to levels below that of the SV40 KO control (<1%) (Fig. 3c, Supplementary Fig. 5). This inhibition may arise from multiple NLS peptides engaging separate nuclear pore complexes for translocation, effectively tethering DNA outside the nuclear envelope, as its length (>1 kbp) exceeds the typical spacing between adjacent pores[27].

To benchmark our system, we adapted the strategy of a previous study by incubating our gene cassettes with NLS peptides (allowing them to electrostatically interact) and assessed eGFP expression following lipofection-mediated transfection into growth-arrested HepG2 cells[36]. Electrostatic interactions allow for multiple NLS peptides to interact with DNA, albeit with little control of the number or location of peptides on a DNA copy. No significant differences in expression ($p > 0.05$) were observed between electrostatically associated NLS-DNA cassettes and controls (Supplementary Fig. 6). Together with the lack of enhancement observed for dual NLS-modified gene cassettes, these results indicate that a single, site-specific NLS modification is optimal for promoting gene expression through improved nuclear translocation.

While these results strongly suggest that NLS-modified DNA cassettes enhance expression by promoting nuclear translocation, they do not directly demonstrate increased nuclear import. To quantify total DNA delivery to growth-arrested HepG2 cells, we lysed cells 48 h post-transfection and performed quantitative PCR (qPCR) to determine absolute DNA copy numbers. There was no significant difference ($p > 0.05$) in total intracellular DNA between NLS-modified and unmodified cassettes, indicating that enhanced gene expression from NLS-modified constructs is not simply due to increased cellular uptake via lipofection (Supplementary Figs. 7, 8). However, qPCR analysis of isolated nuclear fractions revealed a significant ($p < 0.05$) enrichment (~10-20 fold) of NLS-modified DNA compared to controls (Fig. 3d), providing direct evidence that NLS conjugation facilitates nuclear

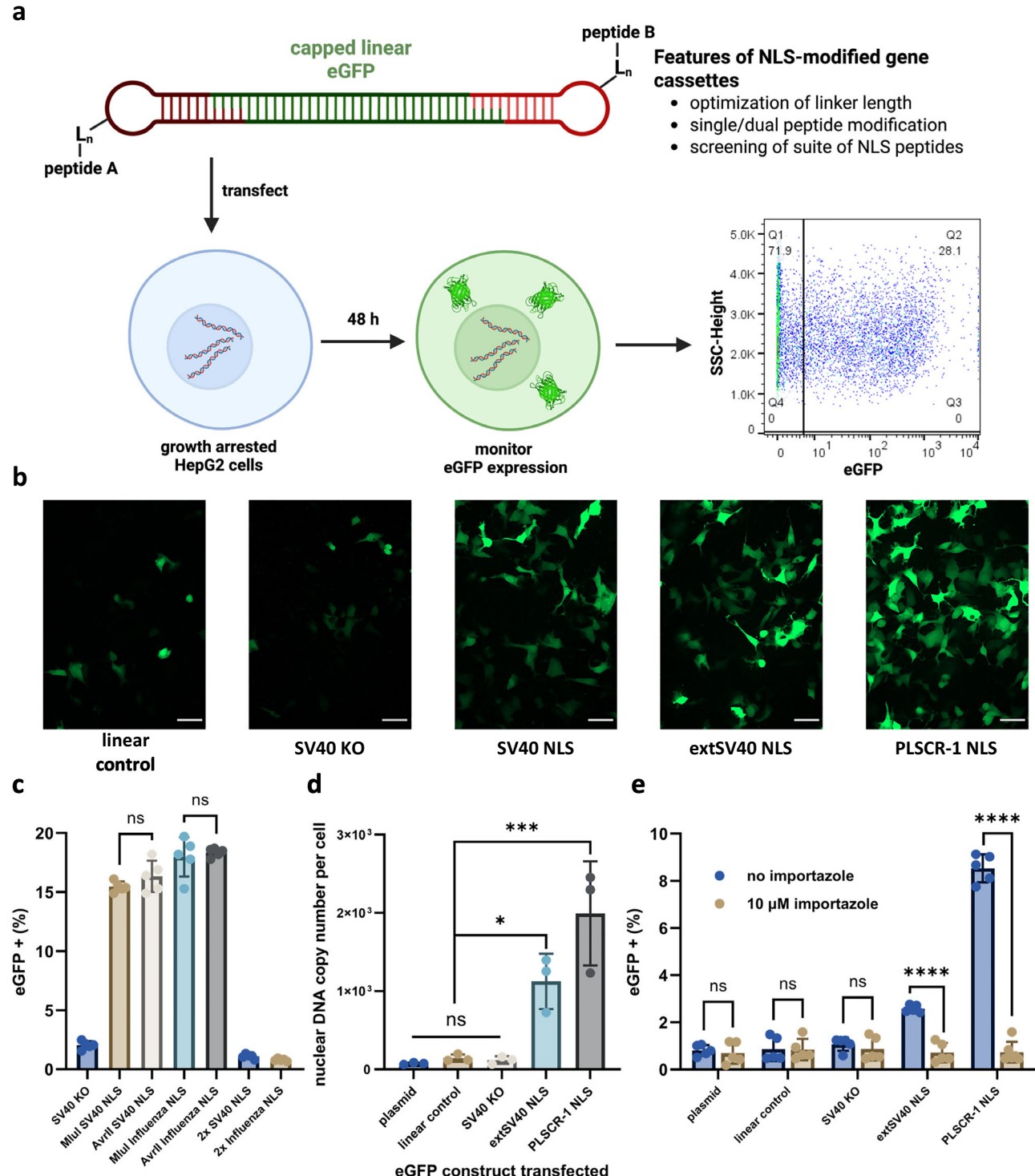

**a**

capped linear eGFP

peptide B

Features of NLS-modified gene cassettes
- optimization of linker length
- single/dual peptide modification
- screening of suite of NLS peptides

peptide A

transfect

growth arrested HepG2 cells

48 h

monitor eGFP expression

**b**

linear control  |  SV40 KO  |  SV40 NLS  |  extSV40 NLS  |  PLSCR-1 NLS

**c** eGFP + (%) — SV40 KO, MluI SV40 NLS, AvrII SV40 NLS, MluI Influenza NLS, AvrII Influenza NLS, 2x SV40 NLS, 2x Influenza NLS

**d** nuclear DNA copy number per cell — plasmid, linear control, SV40 KO, extSV40 NLS, PLSCR-1 NLS; eGFP construct transfected

**e** eGFP + (%) — no importazole / 10 µM importazole — plasmid, linear control, SV40 KO, extSV40 NLS, PLSCR-1 NLS

import of DNA cargo. The observed increase in nuclear copy number, from hundreds to thousands of DNA molecules per nucleus, aligns with previously reported thresholds required for robust gene expression[20,57–60]. This correlation is in agreement with the elevated gene expression we observed for NLS-modified gene cassettes. Consistent with this, eGFP mRNA quantification from transfected cells confirmed that enhanced nuclear accumulation of NLS-modified DNA cassettes corresponds to both higher transcription and subsequent expression (Supplementary Fig. 9). To probe the mechanism by which NLS-modified DNA interacts with nuclear import factors, we transfected importazole-treated HepG2 cells (Fig. 3e). Importazole, an importin-β inhibitor, reduced eGFP expression of NLS-modified

expression cassettes to levels within error of various controls, further supporting NLS-mediated DNA translocation via importin-β[61].

Finally, in the experiments presented here, growth arrest was achieved using aphidicolin, which inhibits DNA polymerase and arrests the cell cycle at the G1/S phase. To assess whether the specific method of growth arrest influences nuclear translocation of DNA cargo, we compared eGFP expression from NLS-modified gene cassettes in HepG2 cells treated with nocodazole, which arrests cells in M phase. Expression levels in nocodazole-treated cells were comparable to those observed in aphidicolin-arrested HepG2 cells (Fig. 4b, Supplementary Fig. 10). In contrast, actively dividing HepG2 cells showed no significant differences (p > 0.05) in eGFP expression between NLS-

**Fig. 3 | Optimization and Nuclear Localization of NLS-modified gene cassettes.**
**a** Parameters of NLS-modified eGFP gene cassettes were optimized: $(GGGGS)_n$ linker length $(L_n)$, single/dual(MluI/AvrII) conjugation, and peptide sequences. Linear eGFP cassettes were transfected into aphidicolin growth arrested HepG2 cells, and eGFP expression was monitored by microscopy and flow cytometry. Created in BioRender. Mohamedshah, Z. (https://BioRender.com/y5bnm15).
**b** Epifluorescence microscopy images showing eGFP expression in growth-arrested HepG2 cells 48 h after transfection with linearized eGFP cassette. Shown are capped, linearized cassettes with no peptide (linear control), an SV40 KO peptide (a single amino acid substitution, K→T, from the SV40 NLS that is known to inhibit nuclear translocation), and three example NLS peptides. "Linear control" is the capped, unmodified linear eGFP cassette. Scale bar: 50 μm. Shown are representative images from 3 independent experiments. **c** Percent of growth-arrested HepG2 cells with eGFP fluorescence 48 h after transfection with NLS-modified DNA cassettes in which the NLS (SV40 or Influenza) is on the 5′ MluI side (peptide A), 3′ AvrII side (peptide B), or both sides (peptides A and B). The "SV40 KO" negative control is the linear cassette modified on the 5′ MluI side with the SV40 KO peptide. All peptides utilize the 3x linker $((GGGGS)_3)$. **d** Copy number of the eGFP DNA gene cassette in the nucleus (per cell) 48 h after lipofection. Peptide modifications are on

the 5′ MluI side of the gene cassette with the 3x linker $((GGGGS)_3)$. Controls include eGFP plasmid, unmodified, capped linear cassette, and the SV40 KO modified cassette. **e** Percent of growth-arrested HepG2 cells with eGFP fluorescence 24 h after transfection with eGFP cassettes with or without 10 μM importazole treatment. Peptide modifications are on the 5′ MluI side of the gene cassette with the 3x linker $((GGGGS)_3)$. Controls include eGFP plasmid, unmodified, capped linear cassette, and the SV40 KO modified cassette. Data collected in (c) and (e) were quantified using flow cytometry and are presented as mean ± s.d. for $n = 5$ biologically independent experiments, with individual data points shown, 100 ng DNA transfected per condition. Data collected in (d) were quantified using qPCR, normalized against RPP30 housekeeping gene, and are presented as mean ± s.d. for $n = 3$ biologically independent experiments, with individual data points shown, 100 ng DNA transfected per condition. For (c) and (d), statistical analysis was performed using one-way ANOVA with Tukey's multiple comparison (*$p \leq 0.05$, **$p \leq 0.01$, ***$p \leq 0.001$, ****$p \leq 0.0001$, ns $p > 0.05$). For (e), statistical analysis was performed using multiple one-way unpaired $t$-tests (*$p \leq 0.05$, **$p \leq 0.01$, ***$p \leq 0.001$, ****$p \leq 0.0001$, ns $p > 0.05$). Source Data are provided as a Source Data file including individual $p$-values.

### Table 1 | Suite of NLS peptides selected for nuclear translocation of DNA cargos

| NLS Peptide | Sequence (N→ C)* | Net Charge (pH 7.4) | Mechanism of Nuclear Import |
|---|---|---|---|
| SV40[31,35,62] | **PKKKRKVEDPYS** $(GGGGS)_3K(N_3)$ | 3+ | Prototypical classical, positively charged NLS Binds to major-NLS binding domain of importin-α; triggering importin mediated nuclear translocation |
| extSV40[31,62] | **SSDEEATADQHSTPPKKKRKVEDPYS** $(GGGGS)_3K(N_3)$ | 1- | Same mechanism as Sv40 NLS Longer sequence has two distinct regions of charged amino acids |
| Influenza A Virus A NP[63,64] | **MASQGTKRSSYEQMETDGERQS**$(GGGGS)_3K(N_3)$ | 1- | Non-classical importin-α NLS Binds to a distinct α-domain independent of major-NLS binding domain of importin-α |
| IGFBP-2[65] | **KHHLGLEEPKKLRPPPAR** $(GGGGS)_3K(N_3)$ | 3+ | IGFBP-2 highly expressed in serum and tumor tissues of many cancers<br>Nuclear import through classical importin-α mediated transport |
| Hrp-1[66] | **RSGGNHRRNGRGGRGGYNRRNNGYHPY** $(GGGGS)_3K(N_3)$ | 7+ | Non-classical proline tyrosine-NLS (PY-NLS)<br>Recognized by the human karyopherin β2/transportin (Kapβ2) nuclear import enzyme |
| PLSCR-1[67] | **GKISKHWTGI** $(GGGGS)_3K(N_3)$ | 2+ | Derived from phospholipid scramblase 1 Hydrophobic residues present with a unique importin-α binding domain |
| Borna Disease Virus p10 (BVP)[68] | **LRLTLLELVRRLNGNG** $(GGGGS)_3K(N_3)$ | 2+ | Non-classical importin-α NLS Hydrophobic residues present |
| HTLV-1 Rex[31,69] | **MPKTRRRPRRSQRKRPPT** $(GGGGS)_3K(N_3)$ | 9+ | Non-classical NLS which binds directly to importin-β95 Basic amino acid sequence rich in arginine |

*Linker sequence $(GGGGS)_3K(N_3)$ is colored grey.

modified gene cassettes and controls (Supplementary Fig. 10). These findings underscore the importance of arresting the cell cycle in isolating NLS-mediated nuclear import, as nuclear envelope breakdown during mitosis can otherwise facilitate passive entry of DNA cargo into the nucleus[17–20].

### Screening NLS peptides for enhanced gene expression

Based on the above results, we used our eGFP expression cassette with a single 5′ MluI modified NLS with a $(GGGGS)_3$ linker for screening alternative NLS peptides for nuclear translocation efficiency. Previous studies have primarily focused on a limited set of NLS sequences, primarily the SV40 NLS, leaving open questions about whether alternative NLS sequences might improve DNA nuclear translocation efficiency[24–31]. We selected the eight NLS peptides indicated in Table 1 and Supplementary Table 2, based on amino acid composition, polarity/charge state, length, mechanism of proposed nuclear translocation, and origin of NLS[31,35,62–69]. Peptides were synthesized by solid phase peptide synthesis including a flexible amino acid linker and azido amino acid at the C-terminus. These modifications to the NLS sequences were included at the C-terminus to minimize disruption of the NLS activity of the peptides while maintaining high yields and purity through

synthesis. All eGFP constructs were lipofected into three growth-arrested human cell lines (HepG2-derived from hepatic cells, HEK293T-embryonic kidney cells, and AC16-derived from primary ventricular tissue) and monitored for eGFP expression by flow cytometry (Fig. 4).

Across all three cell lines, we observed that active NLS modifications of the eGFP gene cassette resulted in greater expression of eGFP compared to negative controls, indicating that the presence of an NLS on a DNA cargo appears to be a generalizable method for improving its nuclear delivery. As previously mentioned, the presence of a peptide alone on the DNA cargo does not appear to be sufficient for nuclear delivery, as the SV40 KO peptide had low eGFP expression levels (comparable to the other controls), consistent with its inability to facilitate nuclear translocation[33,35]. Interestingly, the relative trends in eGFP expression levels among the various NLS peptides were variable across cell types. For example, while the PLSCR-1 ( ~ 5-10 fold increase compared plasmid) and extSV40 NLS ( ~ 3.5-7 fold increase compared to plasmid) modifications consistently facilitated the highest eGFP expression across all three cell types, the other NLS modifications were more variable in their degree of expression enhancement depending on cell type[67]. PLSCR-1 may bind importin-α more effectively than other NLS peptides when conjugated to a DNA cargo due to its short,

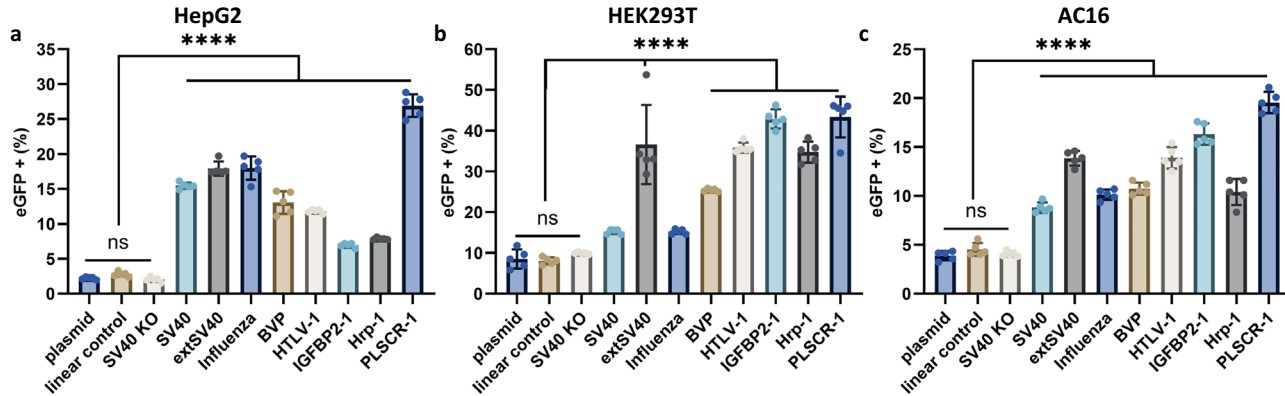

**Fig. 4 | Screening NLS peptides for enhanced gene expression in multiple cell types.** Percent of growth-arrested HepG2 (**a**), HEK293T (**b**), or AC16 cells (**c**) with eGFP fluorescence 48 h after transfection with peptide modified eGFP cassettes. Data collected were quantified using flow cytometry and are presented as mean ± s.d. for $n = 5$ biologically independent experiments, with individual data points shown, 100 ng DNA transfected per condition. Controls include eGFP plasmid DNA, unmodified, capped linear eGFP cassette, and SV40 KO modified eGFP cassette. All peptide modifications were on the 5′ MluI side of the cassette with the 3x linker ((GGGGS)₃). Statistical analysis was performed using one-way ANOVA with Tukey's multiple comparison (*$p \leq 0.05$, **$p \leq 0.01$, ***$p \leq 0.001$, ****$p \leq 0.0001$, ns $p > 0.05$). Source Data are provided as a Source Data file including individual $p$-values.

relatively nonpolar nature, which minimizes electrostatic interactions with DNA. In contrast, previous DNA-NLS conjugates have primarily relied on positively charged NLS peptides, such as the minimal SV40 NLS, which may have resulted in reduced nuclear delivery due to electrostatic interactions with the DNA[28,32,37]. The enhanced nuclear delivery of the extSV40 NLS modification, particularly over SV40, may be due to its pseudo-bipartite nature, where it has two distinct NLS binding domains, one positively charged and one negatively charged resulting in an overall near-neutral charge[31,62]. The observed cell type-specific variability in eGFP expression may also arise from differences in expression profiles of endogenous importin-α isoforms (KPNA1-7), which are known to vary across cell and tissue types including hepatic, renal, and cardiac tissues[70–74]. For example, KPNA1, KPNA2, KPNA6 are known to be expressed in HepG2s, whereas KNPA7 is minimally expressed[75,76]. Such variation in importin-α isoform type and abundance could influence the binding affinities and, consequently, the transport efficiencies of individual NLS peptides conjugated to DNA cargos. However, the specific binding mechanisms between different NLS peptides and importin-α remain under studied, making it difficult to predict NLS preference for a specific cell or tissue type given importin-α isoform expression levels. Additionally, factors such as peptide composition and linker length may further modulate NLS accessibility and nuclear import efficiency in a cell-specific manner that depends on importin-α isoform expression, as both peptide flexibility and local charge environments can impact recognition by importin complexes. Our findings support that specific NLS peptides may be preferred for DNA nuclear translocation in certain cell types, as the IGFBP2-1 and HTLV-1 NLS modifications show lower relative eGFP expression levels in hepatic-derived HepG2 cells, but significantly enhanced eGFP expression in kidney-derived HEK293T and cardiomyocyte AC16 cells. These data demonstrate the potential for designing NLS peptides tailored for selective DNA cargo expression in specific tissues.

### Therapeutic considerations of NLS-modified genes

For therapeutic delivery of DNA, minimizing the required DNA dose is critical to reducing cytotoxicity and immune activation[77]. To evaluate the potential of DNA-NLS conjugates for therapeutic delivery, we transfected growth-arrested HepG2 cells with decreasing amounts of eGFP expressing DNA (Fig. 5a). Based on prior screening (Fig. 4), we selected the PLSCR-1 and extSV40 NLS peptide modifications, as they most effectively enhanced gene expression in HepG2 cells. As expected, reducing the transfected DNA amount (100 ng to 1 ng per well)

decreased overall eGFP expression. However, NLS-modified DNA constructs exhibited significantly higher eGFP expression ($p \leq 0.001$) than all controls at every DNA concentration. Notably, eGFP expression levels when only 10 ng of NLS-modified gene cassette was transfected (~ 5–6%) surpassed that of the controls when 100 ng of DNA was transfected (~ 2-4%). These findings indicate that DNA-NLS conjugates may enhance gene delivery efficiency at low doses, suggesting potential for therapeutic applications.

Prior studies have highlighted the influence of DNA size on translocation, prompting us to evaluate the effectiveness of our approach for various sizes of DNA payloads[28,31,79]. In addition to our original eGFP gene cassette (2537 bp), we generated linear expression cassettes amenable to peptide modification for miniGFP1(1361 bp, a small-sized GFP variant) and ABE8e-P2A-eGFP (6976 bp, a genome editing agent that is co-expressed with eGFP via the ribosomal skipping sequence P2A)[80,81]. We lipofected growth-arrested HepG2 cells with NLS-modified miniGFP1, eGFP, and ABE8e-P2A-eGFP gene cassettes (using either the extSV40 or PLSCR-1 NLS) and evaluated GFP expression (Fig. 5b). We observed statistically significantly ($p \leq 0.001$) enhancement of GFP expression with NLS-modified constructs compared to their matching controls for all three DNA cargos (Fig. 5b). Expression levels were comparable between miniGFP1 and eGFP, while ABE8e-P2A-eGFP exhibited lower expression, which may be due to its ~3-fold larger open reading frame causing constraints on mammalian expression, and/or reduced nuclear translocation efficiency. Nevertheless, NLS-modified gene cassettes can accommodate cargos up to ~7000 bp while maintaining enhanced nuclear delivery.

Finally, we assessed the potential of NLS-modified gene cassettes to express a clinically relevant gene, coagulation *Factor IX*. Factor IX is a secreted serine protease essential for blood coagulation, and mutations in its gene cause hemophilia B (Christmas disease)[82]. Current treatments rely on recombinant factor IX administration, though gene therapy is an emerging alternative[82–84]. In November 2022, the FDA approved Hemgenix, an adeno-associated virus (AAV)-based gene therapy for *Factor IX* expression in the liver. However, due to high costs and safety concerns, there is growing interest in non-viral gene therapy approaches.

We designed a human *Factor IX* gene cassette compatible with NLS peptide modifications, appended each of the eight NLS peptides listed in Table 1 to the 5′ MluI side of the cassette, transfected each construct into growth-arrested HepG2 cells, and measured factor IX secretion via ELISA (Fig. 5c). NLS-modified *Factor IX* cassettes exhibited significantly higher secretion ($p \leq 0.001$) than controls, mirroring the

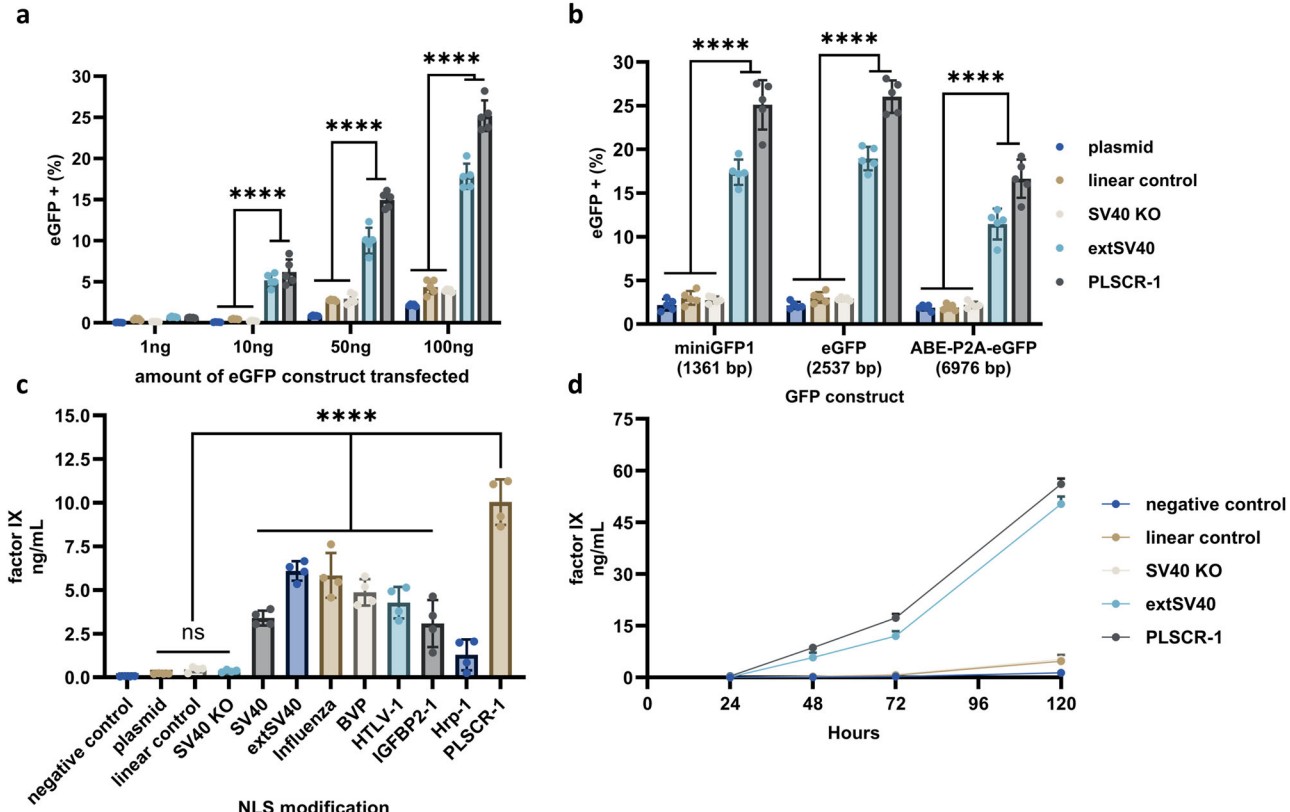

**Fig. 5 | Therapeutic considerations of NLS modified gene cassettes. a** Percent of growth-arrested HepG2 cells with eGFP fluorescence 48 h after transfection with various quantities of NLS-modified eGFP cassettes. **b** Percent of growth-arrested HepG2 cells with eGFP fluorescence 48 h after transfection with various sizes of eGFP-encoding DNA cargos. Data collected in (a) and (b) were quantified using flow cytometry and are presented as mean ± s.d. for *n* = 5 biologically independent experiments, with individual data points shown, 100 ng DNA transfected per condition for (b). Controls include plasmid DNA, unmodified, capped linear cassette, and SV40 KO-modified cassette. All peptide modifications were on the 5′ MluI side of the cassette with the 3x linker ((GGGGS)₃). **c** Concentrations of secreted factor IX protein in serum from growth-arrested HepG2 cells 48 h after transfection with DNA cassettes encoding factor IX. Controls included untransfected cells (labeled as "negative control"), factor IX-encoding plasmid, unmodified, capped linear

cassette, and SV40 KO modified cassette. Data collected in (c) and were obtained by ELISA and presented as a mean ± s.d. for *n* = 4 biologically independent experiments, with individual data points shown, 100 ng DNA transfected per condition. Statistical analysis in (a–c) was performed using one-way ANOVA with Tukey's multiple comparison (*\*p ≤ 0.05, \*\*p ≤ 0.01, \*\*\*p ≤ 0.001, \*\*\*\*p ≤ 0.0001, ns p > 0.05*). **d** Secretion of factor IX from growth-arrested HepG2 cells at time points up to 120 h following transfection of NLS-modified cassettes. Controls include untransfected cells (labeled as "negative control"), unmodified, capped linear cassette, and SV40 KO modified cassette. Data collected in (d) and were obtained by ELISA and presented as a mean ± s.d. for *n* = 3 biologically independent experiments, 100 ng DNA transfected per condition. Source Data are provided as a Source Data file including individual *p*-values.

enhancement seen in eGFP expression (Fig. 4). Notably, the same NLS peptides that enhanced eGFP expression in HepG2 cells also increased factor IX secretion, following identical trends across both gene cassettes within the same cell type (Figs. 4b and 5c). This further supports the notion that specific NLS peptides may facilitate nuclear translocation in a cell- or tissue-dependent manner, regardless of the gene being delivered. We selected the two most effective NLS peptides in this cell line (extSV40 and PLSCR-1) and monitored factor IX secretion over the course of 120 h (Fig. 5d). At time points of 48 h and higher, both NLS-modified constructs maintained significantly higher secretion levels (~10-fold increase at 120 h) compared to controls. Importantly, as media was replaced at each time point without additional DNA delivery, expression beyond 24 h indicates sustained transcription from the initial transfection. These findings demonstrate that NLS-modified gene cassettes can enhance the expression of therapeutically relevant proteins, supporting their future potential as a strategy for both recombinant protein production and non-viral gene therapy applications.

Efficient nuclear delivery of therapeutic DNA remains a key challenge in gene therapy. In this study, we developed a modular platform, DNA-PepTAG, to generate site-specific NLS-modified gene cassettes. Our results demonstrate that gene cassettes modified with a single NLS

peptide show significantly enhanced nuclear localization and expression in growth-arrested cells compared to DNA alone (both plasmid DNA and non-conjugated linear DNA). This is particularly relevant for therapeutic applications targeting non-dividing or slow-dividing tissues, such as the liver, kidney, or muscle, where nuclear delivery is a major bottleneck for exogenous gene expression.

Through systematic screening of NLS modifications, we identified that while NLS conjugation consistently increased expression, the efficacy of specific NLS peptides varied across different cell types, suggesting a level of cell-specific preference. This highlights the importance of tailored nuclear targeting strategies of NLS peptides for different therapeutic contexts. Additionally, we found that a single NLS modification was sufficient to enhance expression across various DNA cargo sizes, whereas dual modifications on opposite ends of the cassette were detrimental to expression. This suggests a finely balanced interaction between nuclear transport efficiency and provides a critical design parameter for future DNA-NLS conjugates.

In this study, DNA–NLS conjugates were delivered via lipofection, a well-established in vitro transfection method. For clinical translation, a natural progression will be to incorporate DNA-NLS conjugates into lipid nanoparticles (LNPs), which represent the leading non-viral delivery modality for nucleic acid therapeutics. The chemical and

structural compatibility of DNA-NLS conjugates with established LNP formulation strategies should be further evaluated to determine if this approach is readily adaptable for in vivo delivery. Future studies will focus on optimizing DNA-NLS LNP formulations and evaluating their efficacy and biodistribution in murine models. Moreover, expanding the screening of NLS peptides from diverse sources may allow for more refined nuclear-targeting strategies, particularly for tissue- or cell-type-specific delivery. The ability to rationally or synthetically design novel peptides tailored for DNA transport further enhances the potential of this approach. Beyond NLS modification, this platform is broadly adaptable, allowing for the conjugation of DNA with other peptides or azide-containing moieties, expanding its potential for diverse nucleic acid delivery applications.

Overall, our findings establish a robust strategy for enhancing nuclear delivery of DNA therapeutics. This work not only lays the foundation for further development of peptide-modified DNA systems but also opens avenues for optimizing non-viral gene delivery across a range of clinical applications.

## Methods

### General information

All reagents and buffers were obtained from Sigma-Aldrich unless otherwise specified. Oligonucleotides were synthesized by Integrated DNA Technologies (Supplementary Table 1). DBCO-NHS was purchased from BroadPharm, and preQ$_1$-biotin was provided by Tocris Bioscience.

Polyacrylamide gel electrophoresis (PAGE) was performed using 19:1 sequencing-grade urea PAGE gels (Sequagel system, National Diagnostics) in a Bio-Rad Mini-Protean system. Gels were stained with GelRed (Biotium) and imaged using a Bio-Rad ChemiDoc-MP gel imager. Image analysis was conducted using Fiji/ImageJ software (NIH).

Protein purification was carried out via fast protein liquid chromatography (FPLC) using a Cytiva ÄKTA Pure system. All natural Fmoc-protected amino acids were obtained from CEM, Chem-Impex Int'l, ChemPep, Sigma-Aldrich, and TCI. Fmoc-L-Lys(N$_3$)-OH was purchased from Sigma-Aldrich. Peptide synthesis was performed at a 0.1 mmol scale using a CEM Liberty Blue Microwave Peptide Synthesizer. Trityl-OH ChemMatrix resin (100–200 mesh, 0.37 mmol/g) from Biotage was used for microwave-assisted solid-phase peptide synthesis.

Flow cytometry experiments for GFP expression were conducted on a Bio-Rad S3e Cell Sorter. Confocal microscopy was performed using a Zeiss Cell Observer® SD spinning disk system (Yokogawa, Japan) integrated into an Axio Observer Z1 motorized inverted microscope (Zeiss, Germany). Microscopy images were analyzed using Fiji/ImageJ software (NIH).

All bio-reagents, restriction enzymes, DNA purification kits, and competent bacterial strains were obtained from New England Biolabs, Life Technologies, or Promega. Absorbance measurements were recorded using a Thermo Scientific NanoDrop 2000c UV-Vis spectrophotometer. High-resolution mass spectrometry (HRMS) for small molecule probes, peptides, and modified oligonucleotides was performed using an Agilent 6230 time-of-flight mass spectrometer (TOF-MS) with a JetStream electrospray ionization (ESI) source at the UCSD Department of Chemistry and Biochemistry Molecular Mass Spectrometry Facility (Supplementary Figs. 16–33). Agilent MassHunter workstation (version 10.1) was used for HRMS data acquisition and analysis, and MagTran software was utilized for mass spectrum deconvolution.

Cartoons in Figs. 1, 2d, and 3a were created in Biorender.

### TGT expression and purification

E. coli TGT was expressed as previously described (Supplementary Method 1)[43,45]. Briefly, a plasmid encoding TGT was transformed into BL21(DE3)pLysS chemically competent E. coli (Invitrogen C606010) and plated on LB agar containing kanamycin. Individual colonies were

expanded in a 10 mL overnight culture, then transferred to 1 L of LB with kanamycin. Cells were grown to an optical density of 0.4−0.6, cooled at 4 °C for 1 h, and induced with 1 mM IPTG at 18 °C overnight. Bacterial pellets were collected by centrifugation (6000 × g, 20 min, 4 °C), resuspended in 10 mL lysis buffer (20 mM sodium phosphate, 500 mM NaCl, 20 mM imidazole, 400 µM PMSF, pH 7.7), and sonicated on ice (20 min total, 1 min on/off cycles, 50% duty cycle). The lysate was cleared by centrifugation (10,000 × g, 30 min, 4 °C).

TGT was purified using an ÄKTA pure FPLC system. Crude lysate was loaded onto a HisTrap FF crude column (Cytiva), washed with lysis buffer, and eluted with His elution buffer (20 mM sodium phosphate, 500 mM NaCl, 500 mM imidazole, 400 µM PMSF, pH 7.7). The eluate was further purified by size-exclusion chromatography (SEC) on a Superdex 75 10/300 GL column using TGT storage buffer (25 mM HEPES, 150 mM NaCl, 1 mM EDTA). Enzyme concentrations were determined by A280 absorbance, and aliquots were stored at −80 °C.

### TGT labeling

TGT-mediated DNA modification was performed as previously described[43]. Briefly, reactions were conducted at 37 °C for 4 h in a thermocycler with a heated lid using the following conditions: 10 µM TGT, 2.5 µM DNA oligonucleotide, 20 µM preQ$_1$-X (preQ$_1$-biotin or DBCO), and 1× TGT reaction buffer. Reaction components were mixed, vortexed gently, spun down, and incubated. Samples were analyzed directly via Urea-PAGE without purification, which did not impact gel shift resolution. Gels were analyzed using ImageJ software to determine percent yield by band intensity. PreQ$_1$-DBCO synthesis is described in Supplementary Fig. 34.

### Urea-PAGE analysis

Urea-PAGE was used to assess TGT activity on DNA substrates[43]. Gels were prepared using the SequaGel UreaGel System (National Diagnostics EC-833) and run in a Mini-PROTEAN Tetra Vertical Electrophoresis Cell (Bio-Rad). Fresh gels (10 mL) were polymerized with 14 µL TEMED and 80 µL 10% ammonium persulfate. Samples (150–200 ng DNA oligonucleotide) were diluted in 2× RNA loading dye (NEB B0363S), denatured at 98 °C for 5 min, and loaded into the gel. Electrophoresis was performed at 210 V for 90 min in 1× TBE buffer. Gels were stained with GelRed (1:1000 in TBE) for 5 min and imaged using a Bio-Rad ChemiDoc system. Uncropped, unprocessed scans of presented gels are found in the Source Data.

### Peptide synthesis

Trityl chloride ChemMatrix resin was freshly prepared from trityl-OH ChemMatrix resin (270 mg, 0.1 mmol, 0.37 mmol/g loading). The resin was swelled in DCM/toluene (1:1, 5 mL) for 10 min, treated with AcCl (0.2 mL), and shaken overnight. The resin was washed with DCM (3×10 mL) before coupling with Fmoc-L-Lys(N$_3$)-OH (80 mg, 0.2 mmol) in DCM containing DIEA (140 µL, 0.8 mmol) for 2 h. Capping was performed using DCM:MeOH:DIEA (8.5:1:0.5, 5 mL). Resin loading was quantified by Fmoc deprotection and absorbance measurement at 301 nm.

Peptides were synthesized at a 0.1 mmol scale using a CEM Liberty Blue microwave synthesizer with Fmoc-L-Lys(N$_3$)-ChemMatrix resin. Deprotection was performed with 20% 4-methylpiperidine in DMF. Coupling reactions utilized a 5-fold excess of Fmoc-AA(PG)-OH, 0.5 M DIC in DMF, and 1.0 M Oxyma Pure (0.1 M DIEA in DMF). Following synthesis, peptides were cleaved using a TFA:TIS:H$_2$O:DODT (92.5:2.5:2.5:2.5) cocktail for 3 h. The filtrate was evaporated under argon, and peptides were precipitated in cold Et$_2$O, centrifuged (7000 x g, 5 min), washed with Et$_2$O, and dried under vacuum.

Purification was performed via HPLC using a Zorbax SB-C18 semipreparative column with Phase A (H$_2$O + 0.1% TFA) and Phase B (CH$_3$CN + 0.1% TFA) gradients. Analytical HPLC was conducted on an Eclipse Plus C8 column with Phase A (H$_2$O + 0.1% formic acid) and

Phase B ($CH_3CN$ + 0.1% formic acid). HRMS was used to verify peptides (Supplementary Figs. 16–30).

## Copper-free click reactions (SPAAC) for DNA oligonucleotide-peptide conjugation

DBCO-modified DNA oligonucleotides were conjugated to azido-peptides via strain-promoted azide-alkyne click (SPAAC) chemistry (Supplementary Fig. 35). SPAAC enables efficient triazole linkage formation between a strained cyclooctyne (DBCO) and an azide with near-quantitative yields under simple aqueous conditions. Following the TGT labeling reaction to modify DNA oligonucleotides with DBCO, azido-peptides were added directly to the reaction tube at a final concentration of 100 μM (5:1 azide to DBCO). The reaction was incubated at 37 °C overnight with gentle shaking. DNA oligonucleotide-peptide conjugates were analyzed via UREA-PAGE and purified using the Oligo Clean & Concentrator kit (Zymo Research, D4061), eluting in pure water for subsequent ligation to the gene cassette or for HRMS analysis (Supplementary Figs. 31–33).

## Generation of NLS-Modified Gene Cassettes

The coding sequence of pcDNA 3.1 (+) eGFP was PCR-amplified using NEB Q5 High-Fidelity DNA Polymerase (NEB, M0494L). PCR products were purified with the Monarch Spin PCR & DNA Cleanup Kit (NEB, T1130L), eluted in pure water, and sequence-verified by Plasmidsaurus. To generate sticky-end gene cassettes, the PCR product underwent dual restriction digestion with MluI-HF (NEB, R3198L) and AvrII (NEB, R0174L) for 1 h at 37 °C (Supplementary Method 1). Digested DNA was purified using the Monarch Spin kit by NEB.

Sticky-end DNA fragments were ligated to corresponding DNA oligonucleotides (5:1 oligonucleotide to DNA) overnight at 16 °C using T4 DNA ligase (NEB, M0202L). The ligated DNA oligonucleotides were optionally modified with peptides as described earlier. Purified ligated linear DNA underwent T7 exonuclease (NEB, M0263S) treatment for 30 min at 37 °C to remove unligated DNA. Successful ligation was confirmed via agarose gel electrophoresis (Supplementary Fig. 3) and sequencing by Eton Biosciences. The same approach was used to generate NLS modified miniGFP1 and factor IX gene cassettes (Supplementary Method 1). For ABE8e-P2A-eGFP gene cassettes, a similar strategy was applied using AflIII (NEB, R0541S) instead of AvrII, with an AflIII sticky-end DNA oligonucleotide for ligation (Supplementary Method 1).

## Cell culture

HepG2 (HB-8056, ATCC), HEK293T (CRL-3216, ATCC), and AC16 (CRL-3568, ATCC) cells were cultured per ATCC guidelines. HepG2 and HEK293T cells were maintained at 37 °C with 5% $CO_2$ in complete growth medium containing Dulbecco's modified Eagle medium (DMEM, Gibco 11995065), 10% FBS, and 1% penicillin/streptomycin (Gibco 15070063). AC16 cells were cultured under similar conditions in DMEM/F12 (Thermo Fisher 11320-033) supplemented with 12.5% FBS and 1% penicillin/streptomycin.

For experiments, cells were seeded at ~50% confluency in collagen-coated 96-well plates and incubated for 48 h. Cells were then washed with PBS (pH 7.4) and cultured in complete medium supplemented with 0.75 μM aphidicolin (Thermo Fisher J60236.MCR) to induce growth-arrest in S-phase. Cells remained in aphidicolin-containing media for 48 h, after which growth-arrest was confirmed using a Click-iT EdU Proliferation Assay (Invitrogen C10499). Growth-arrested cells were subsequently transfected with eGFP or factor IX DNA constructs. For nocodazole growth arrest the same method was used with complete media supplemented with 200 ng/mL nocodazole (Thermo Scientific).

## Flow cytometry for eGFP expression quantification

HepG2, HEK293T, and AC16 cells were seeded in 96-well plates and transfected following 48 h of growth-arrest. DNA mixtures were diluted in Opti-MEM reduced-serum medium (Gibco, 31985062) to a final volume of 5 μL and combined with 0.3 μL of Lipofectamine 3000 (Invitrogen, L3000008) diluted to 5 μL in Opti-MEM, following manufacturer instructions. Notably, the P3000 reagent was excluded to prevent interference from NLS-containing lipid cations that could impact nuclear translocation observations.

Unless otherwise specified (e.g., for DNA titration experiments in Fig. 5a), 100 ng of plasmid, linearized gene cassette, or peptide-modified gene cassette was delivered per well. After 48 h, cells were washed with 100 μL PBS and detached with 50 μL (20 μL for HEK293T cells) of Accumax (STEMCELL Technologies) for 5 min at 37 °C. Cells were then resuspended in 100 μL of cold sort buffer (1 mM EDTA, 2% FBS in PBS), passed through a 35 μm cell strainer, and maintained on ice. Samples were analyzed using an S3e Cell Sorter (BioRad) to quantify eGFP-expressing cells. Example gating strategies for scatter and fluorescence are shown in Supplementary Figs. 11, 12, and 13, with controls for both (-) and (+) eGFP populations.

## Quantitative PCR for DNA/RNA quantification

For determination of total transfected DNA copy numbers, HepG2 cells were seeded in 96-well plates and transfected following 48 h of growth-arrest as described above. After 48 h, cells were washing 100 μL PBS and lysed with 50 μL lysis buffer (10 mM Tris–HCl, pH 8.0, 0.05% SDS, 25 μg/mL Proteinase K in water). Cells were allowed to incubate for 1 h at 37 °C, and lysate was used as template for qPCR (1:20 lysate dilution).

For determination of nuclear and extranuclear DNA copy numbers, HepG2 cells were seeded in 96-well plates and transfected following 48 h of growth-arrest as described above. After 48 h, cells were detached with 50 μL TripLE (Gibco, 12604013). Cells were harvested, centrifuged (300 x g, 5 min, 4 °C), washed with PBS, and centrifuged again. The resulting cell pellet was resuspended in cold cytosolic extraction buffer (10 mM Tris-HCl, pH 7.5, 10 mM NaCl, 3 mM $MgCl_2$, 250 mM sucrose, and 0.02% Digitonin in water)and incubated on ice for 5 min, swirling every minute. Samples were centrifuged (300 x g, 5 min, 4 °C), with the resulting pellet containing intact nuclei and the supernatant containing the extranuclear fraction. The nuclear fraction and extranuclear fractions were separated and subjected to lysis as described above and used as template for qPCR (1:20 dilution).

qPCR to determine DNA copies were obtained using PerfeCTa™ SYBR® Green FastMix™, Low ROX™ kit (Quantabio) on a Bio-Rad CFX Connect Real-Time PCR Detection System. eGFP DNA was amplified with primers reading through the CMV region (FW- GCAGTA-CATCAATGGGCGTG, RV-GTCCCGTTGATTTTGGTGCC). Resulting Cq scores were normalized against the RPP30 gene and quantified by a calibration curve[85].

For determination of mRNA transcripts, HepG2 cells were seeded in 96-well plates and transfected following 48 h of growth-arrest as described above. After 48 h, cells were washing 100 μL PBS and lysed with 50 μL lysis buffer (10 mM Tris–HCl, pH 8.0, 0.05% SDS, 25 μg/mL Proteinase K, RNase inhibitor (murine (NEB) in water). Cells were allowed to incubate for 1 h at 37 °C, and lysate was used as template for qPCR (1:20 lysate dilution).

RT-qPCR was performed using the Luna® Universal One-Step RT-qPCR Kit (NEB, E3005) to determine mRNA transcripts on a Bio-Rad CFX Real-Time PCR Detection System. eGFP mRNA was amplified with the following primers: FW – AGGACGACGGCAACTACAAG, RV – AAGTCGATGCCCTTCAGCTC. Resulting Cq scores were normalized against GAPDH transcription and quantified by a calibration curve.

## Importazole assay for importin-β inhibition

To determine optimal importazole (SML0341, Sigma-Aldrich) concentrations for importin-β inhibition, growth-arrested HepG2 cells were transfected with a NLS-eGFP plasmid and treated with increasing concentrations of importazole (0–25 μM) based on prior studies[61,86]. After 24 h, eGFP localization was monitored by fluorescence imaging

to assess cytosolic retention, indicative of importin-β inhibition (Supplementary Fig. 14). A concentration of 10 µM was identified as optimal, effectively inhibiting importin-β while maintaining cell viability. Growth-arrested HepG2 cells were then treated with 10 µM importazole, transfected with eGFP constructs, and analyzed for eGFP expression by flow cytometry after 24 h.

### Factor IX ELISA

HepG2 cells were seeded in 96-well plates and transfected following 48 h of growth-arrest. DNA mixtures were prepared as described earlier, using 100 ng of factor IX plasmid, linearized gene cassette, or peptide-modified gene cassette per well. After 48 h, secreted factor IX levels were measured using an ELISA kit (ELH-F9, RayBiotech) according to the manufacturer's instructions. For the 48-h expression study, 100 µL of media was collected from each well for factor IX quantification. In the time-course study, 100 µL of media was collected at each time point and replaced with 100 µL of fresh growth-arrest media until the next collection point.

### Statistics and reproducibility

Statistical analysis was done using GraphPad Prism 10 or Microsoft Excel (Version 2510). All relevant information on statistical analysis, including $p$-values, the number of samples that were analyzed and the statistical test used can be found in the respective figure legends. Individual data values and specific $p$-value comparisons are provided in the Source Data. No statistical method was used to predetermine sample size. No data were excluded from the analyses and experiments were not randomized. The investigators were not blinded to allocation during experiments and outcome assessment.

### Reporting summary

Further information on research design is available in the Nature Portfolio Reporting Summary linked to this article.

## Data availability

All data supporting the findings in this article is provided in the main text, Supplementary Information, or Source Data. Sequences of linear gene cassettes generated in this work have been provided in Supplementary Data 1 and deposited to GenBank (NCBI NLM) under accession codes: PV533970, PV533971, PV533972, PV533973. Source data are provided with this paper.

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

## Acknowledgements

The authors acknowledge financial support from Seawolf Therapeutics, The Camille & Henry Dreyfus Foundation (ST-25-025, to N.K.D.), and the National Institutes of Health (R35GM141939, to N.K.D. and R35GM138317, to A.C.K.). We also acknowledge the UCSD Molecular Mass Spectrometry Facility (MMSF) for HRMS analyses. Z.Y.M. was supported by the Chemistry-Biology Interface Training Program, NIH Grant T32GM146648. We thank Elizabeth Alcamo, Rob Burke, Sataree Khuansuwan, and Oliver Dansereau of Seawolf Therapeutics for helpful discussions and advice throughout the duration of this project. We thank Chuxuan Ling for her help in peptide synthesis.

## Author contributions

Conceptualization: Z.Y.M. and N.K.D. Investigation: Z.Y.M., C. C., E. M. T. Funding acquisition: Z.Y.M. and N.K.D. Supervision: A.C.K. and N.K.D. Writing: Z.Y.M., A.C.K., N.K.D. All authors commented on the manuscript.

## Competing interests

Z.Y.M. and N.K.D. are listed as inventors in a Regents of the University of California provisional US patent (63/802,258), related to the development of DNA-PepTAG and increased nuclear translocation of DNA gene cassettes. N.K.D has previously provided consulting services to Seawolf Therapeutics Inc. A.C.K. is a member of the SAB of Pairwise Plants, is an equity holder for Pairwise Plants and Beam Therapeutics, and receives royalties from Pairwise Plants, Beam Therapeutics and Editas Medicine via patents licensed from Harvard University. N.K.D.'s and A.C.K.'s interests have been reviewed and approved by the University of California, San Diego, in accordance with its conflict-of-interest policies. The remaining authors declare no competing interests.
