## [Peer Review File · Nature Communications]

Highly Efficient Expression of DNA-peptide Conjugates in Growth-arrested Cells

Corresponding Author: Professor Neal Devaraj

Version 0:

Reviewer comments:

Reviewer #1

(Remarks to the Author)

The manuscript of Mohamedshah et al describes a new method to deliver genes to cellular nuclei in order to induce expression of encoded proteins. Using their earlier developed method to replace a guanine with a 7-deazaguanine derivative, they changed it into an approach that incorporates the strained alkyne DBCO into the module. Subsequently, nuclear localization signal (NLS) peptide sequences are attached, after which the genes are introduced into the module. By means of various experiments they show this approach allows the efficient delivery of a gene for the reporter system eGFP to the nucleus of human cells. Although much of the work focusses on HepG2 cells, two other cell lines were also used, illustrating the wider applicability of their approach. The approach was optimized along various axes, including linker length and type of NLS, and evidence is provided that a single mutation in the peptide sequence suppresses gene delivery, as was inferred from the absence of eGFP expression. Importantly, GFP-encoding genes were adequately delivered by this approach, indicating its suitability for the delivery of 7000 bp long genes.

All things considering it is a thoroughly performed study that has sufficient merit to be considered for publication in Nature Communications, provided the following points are addressed and the manuscript is properly revised.

- 1) A list of abbreviations needs to be included.
- 2) When the different set of NLS peptides is provided (e.g., in Figure 4a), please include a column with net charge of the peptide at pH 7.4 (using 1+ for H, K, R, and N-terminus; and 1- for D, E and C-terminus). While updating the table in Figure 4a, I suggest that the linker is displayed in a different font color (e.g., in grey). The authors should also mention why they did not simply acylate the N-terminus with an azide-containing group (which is the most direct approach), but followed a slightly less usual route using azidolysine at the C-terminus. There is probably a good reason for this, and the reader should be informed.
- 3) At the end of the section entitled 'Screening NLS peptides for enhanced gene expression', the results are tentatively connected to importin-alpha isoforms. What are the expression levels of these isoforms in the different cell lines that were used? Also, it is very likely that also linker-length (and composition!) affect transfection in different cell types. Therefore, I encourage the authors to expand this discussion with a few sentences to at least mention all variables that might be at play.
- 4) The observation that the dual-end modified constructs lead to expression levels below the Sv40 KO negative control is very intriguing. If it is indeed the case that the peptides engage two different nuclear pore complexes, essentially glueing it to the outside of the nuclear envelope, this effect might be countered by including shorter linker, i.e., a single GGGGS linker. It would be interesting to see the result of such an experiment, assuming this can be performed conveniently.
- 5) Interestingly, the authors find that constructs that contain two NLS peptides do not lead to efficient eGFP production. This directly relates to a comment in the introduction of multiple NLS peptides per DNA cargo, which states that 'non-site-specific modification that can lead to multiple NLS peptides per DNA cargo.', and I encourage the authors to relate to this when this is treated in the text (e.g., just above the sub-heading 'Screening NLS peptides for enhanced gene expression').
- 6) In their design, the authors also screened various linker lengths. They observe that shorter linkers lead to less efficient eGFP expression than longer linkers, and they attribute it to a better charge separation. How does an increased linker length lead to a better charge separation? One could also argue that a longer spacer leads to more flexibility in the connection between peptide and DNA, which might actually facilitate charge interactions.
- 7) The paper solely relies on Urea-PAGE for purity analysis of the obtained constructs. This type of analysis is heavily biased to the visualization of DNA (GelRed is a nucleic acid staining), and neglects the peptides. Although I have no reason to doubt that the proper constructs have been formed, I strongly encourage the authors to include several reversed-phase HPLC chromatograms of the prepared DNA-NLS conjugates, i.e., at least of the most-used constructs and the mutated

negative control.

8) The authors mention that the DNA-NLS conjugates were obtained in 90-95% yield. However, this is based on DNA, and not on the peptide (the latter is used in 5-fold excess). Also, this is based on the assumption the staining of the DNA-NLS conjugates is not affected by the presence of the peptide, whereas this is not necessarily the case. Evidence that GelRed staining works equally well on DNA and on DNA-NLS conjugates should be provided (could also be a literature reference to a dedicated study on this aspect) before this yield percentage can be reported. Alternatively, another analytical technique needs to be applied for quantitative analysis.

9) Why do the ESI-TOFMS spectra of the Sv40 NLS peptides with linkers (G4S)_n (where $n > 1$) show peaks corresponding to shorter lengths? Specifically Sv40 NLS Lx2 shows peaks from Sv40 NLS Lx1 and Sv40 NLS Lx3 shows peaks of both Sv40 NLS Lx2 and of Sv40 NLS Lx1. Along similar lines, the spectrum for Sv40 NLS Lx4 shows a peak assigned as [M+H+2Na]³⁺ at 972.7300, which could equally originate from Sv40 NLS Lx1 [M+2H]²⁺ (which appears at 973.0276 in the first spectrum and tentatively at 972.5192 in the third spectrum). Similar observations can be made for ESI-TOFMS spectra of most other NLS-peptides in which linker-lengths are varied. I assume these result from anomalies in the analysis, and do not originate from the samples themselves (i.e., the applied conjugates are unique and clean, and do not contain a mixture of components). If, however, evidence of this cannot be provided, it is hard to see how the presented results originate from the indicated conjugates.

10) Change Fmoc-AA-OH into Fmoc-AA(PG)-OH to include the protecting group that is also needed for several amino acids. Additional comments on the figures.

Figure 1, panel a: three different strategies are shown, but they are not clearly separated. Please add visual aids to separate the approaches. The text at the bottom describes that the DNA-NLS conjugate is expressed, but that is not the case. The gene that is carried inside by the DNA-NLS conjugate is expressed. Please rephrase to increase accuracy. Also, the arrow that shows passage of the NPC can be made bolder (also in panel b).

Figure 2, caption panel b: the right part of panel b does not show 'their covalent attachment to a DNA stem loop by TGT', but 'PAGE-urea gel analysis of the covalent attachment to a DNA stem loop by TGT'. Please rephrase, and rephrase 'Urea-PAGE analysis shows an' in the last sentence of this part with 'The'.

Figure 3. The list of features in panel a follows a different order than given in the caption. Please align. Also, a comment should be provided on the origin of the background in Fig3b negative controls?

Figure 4. Statistical analysis is provided on the bars with the largest differences. I think this should also be done between bars with the smallest differences, so that the reader understands the range of differences.

Figure 5. The end of the caption refers to details on how data for the results in panel (c) was obtained. This should move to the caption directly associated to panel (c), and not placed at panel (d).

Reviewer #2

(Remarks to the Author)

MAJOR REVISION RECOMMENDED

In their manuscript entitled "Highly Efficient Expression of DNA-peptide Conjugates in Growth-arrested Cells", Zulfiqar Y. Mohamedshah et al. present a procedure for efficiently obtaining DNA-peptide assemblies designed to overcome the nuclear membrane barrier, which is achieved through nuclear localization signal (NLS) mediating active nuclear import in non-dividing cells. According to the authors, the proposed workflow underscores the potential of DNA-NLS conjugates for non-viral gene therapy applications.

While this study appears both innovative and engaging, I have several comments and questions – primarily concerning the experimental aspects – that require authors' attention. To better convince the reader and enhance the study impact, I think additional investigations and analyses are necessary. Consequently, I recommend a major revision to address the comments outlined below.

Main comments:

A key message of this paper is "enabling enhanced nuclear delivery of therapeutic genes while minimizing the required DNA dose". The potential of the approach is demonstrated through the expression of eGFP or Factor IX cassettes carried by the delivered DNA cargos, both serving as reporter systems i.e., systems reporting the efficiency of the trafficking into the nucleus. While the results offer valuable insights, I think they do not provide sufficient direct evidence of enhanced nuclear import. A more definitive demonstration could be achieved by comparing the copy number of DNA-peptide conjugates in the nucleus with that in the cytoplasm, across different experimental conditions. This could involve subcellular fractionation, quantitative PCR, immunofluorescence, or other relevant methods.

The authors state "the use of growth-arrested cells allows direct assessment of the efficacy of NLS peptides in facilitating nuclear entry". Growth arrest was achieved by inhibiting cell division with aphidicolin. For certain assays, importazole was additionally used to inhibit importin- β , thereby blocking nuclear import. Additionally, it is mentioned in SI "Expression was assessed only at 24 h and not longer due to the cytotoxic properties of importazole.". This raises questions regarding these drugs and their possible direct or indirect effects on the reported results. Exploring other cell division inhibitors, such as nocodazole or colchicine, and comparing results with non-growth-arrested cells could provide complementary valuable insights.

Figure 1 presents a comparison of DNA-NLS conjugates, emphasizing the limitations of previous systems. To better appreciate the results reported and further substantiate the claimed superiority of the approach presented in this study, it would have been valuable to include comparative results of some previously reported DNA-NLS conjugates tested in

parallel.

The authors draw conclusions and claim the “generability” of their approach on the basis of results obtained from three human cell lines. I wonder about its broader applicability, particularly in primary cells, which could offer further valuable insights into nuclear import efficiency under more physiologically relevant conditions.

In most experiments, the transfection efficiency of the various DNA-peptide conjugates was assessed by FACS analysis using an eGFP reporter system, where its expression served as a proxy for successful transfection and readout to monitor differences. While the percentage of positively transfected cells can indeed provide a measure of transfection efficiency, particular caution is warranted when using GFP as a marker. Considering Supplementary Figures 4–6, clarification of the gating strategy used to distinguish between GFP-negative and GFP-positive cells is needed. A dual-gate fluorescence analysis (FL1-H vs. FL2-H) is recommended to account for the inherent autofluorescence of cells and its potential variations, especially following transfection with cationic lipids. Such an approach would help exclude contributions from artifacts, for instance alterations in natural endogenous autofluorescence induced by the transfection reagent itself (see for examples, DOI: 10.2144/01312st02 and DOI: 10.1006/abio.2000.5006). Additionally, incorporating an alternative reporter system such as luciferase would help consolidate and validate the findings further.

Other comments:

The abstract mentions non-viral gene therapy; however, the transfection reagent used, i.e., Lipofectamine 3000, is referenced only once in the manuscript, in the Methods section (page 22). To ensure clarity and coherence, its role should be highlighted more prominently, starting from the abstract. Additionally, the study could be completed with some characterizations of the lipoplexes formed by Lipofectamine 3000 and the various DNA-peptide conjugates. Parameters such as size and charge, as well as results of DNA retardation assays, could be incorporated to provide a more comprehensive analysis.

Regarding the claimed “generability” of DNA-peptide conjugates and their “potential for diverse nucleic acid delivery applications,” the reader may question the effect of alternative gene transfer agents or other delivery methods, such as electroporation.

Toxicity is mentioned some times in the paper, on pages 2 and 12. The authors emphasize the importance to reduce the quantity of transgene to limit any possible side-effect. On page 23, it can be read “A concentration of 10 μ M was identified as optimal, effectively inhibiting importin- β while maintaining cell viability” and in Supplementary Information, it is reported “Among the tested concentrations, 10 μ M importazole exhibited the strongest importin- β inhibition while maintaining cell viability. Higher concentrations (20 and 25 μ M) resulted in increased cytotoxicity.”. However, no supporting data related to cell viability or cytotoxicity is provided.

While the authors evaluated the knockout of SV40, I wonder why a KO of PLSCR-1 was not investigated, given that the latter yielded the best results across most conditions.

Some sentences should be rephrased for more clarity. For instance: “cytosolic and secreted DNA payloads” in the abstract sounds confusing.

What are Q3 and Q4 in Supplementary Figures 4 and 6?

Reviewer #3

(Remarks to the Author)

Delivery of DNA to the nucleus of non-proliferating cells is critical for gene expression but has been challenging in gene therapy. This manuscript reported a new strategy to increase the efficiency by generating DNA oligonucleotide-peptide conjugates. Using a GFP reporter, they optimized the design and screened multiple NLS peptides in growth-arrested human cell lines. They also evaluated the generality of the approach using other gene cassettes. This study provides a new strategy for non-viral gene therapy. However, more evidence is required to support the findings.

1. The major concern is that the entire study relies solely on measuring protein product levels. Direct evidence is needed to quantify the distribution of DNA between the nucleus and cytoplasm. Additionally, the total level of transfected DNA in cells has not been quantified, making it important to rule out the possibility that different constructs affect transfection efficiency. Finally, RNA levels should be measured to confirm that enhanced nuclear import of DNA correlates with increased transcription.

2. The description and diagram of the method are not clear. How is the addition of single versus dual peptides to the gene cassettes controlled?

3. It is not clear whether the linear control DNA contains the end loops. Again it is best to have a diagram of the individual construct used in the paper. And Figure 1 (especially 1a) is more suitable for supplemental information.

4. It is unclear how this approach could be applied clinically—does it require transfection of DNA-peptide conjugates?

Version 1:

Reviewer comments:

Reviewer #1

(Remarks to the Author)

I am satisfied with the changes made by the authors, and congratulate them on their work.

Reviewer #2

(Remarks to the Author)

MINOR REVISION RECOMMENDED

In their revised manuscript entitled "Highly Efficient Expression of DNA-peptide Conjugates in Growth-arrested Cells", Zulfiqar Y. Mohamedshah et al. have addressed most of my comments:

Importantly, the authors conducted comprehensive complementary experiments using quantitative PCR to provide more direct evidence of active nuclear transport. The findings demonstrate that NLS-modified gene cassettes indeed exhibit significantly enhanced nuclear copy number relative to control constructs.

As also requested, eGFP expression from NLS-modified gene cassettes in both nocodazole-arrested and actively dividing HepG2 cells was assessed. The results show that specific growth arrest agent does not affect nuclear translocation of DNA cargo and growth arrest is essential for isolating and assessing NLS-mediated nuclear import.

In addition, DNA-NLS conjugates were compared with those formed through nonspecific electrostatic interactions. The results clearly demonstrate that enhanced gene expression via nuclear translocation requires a single, site-specific, and stable covalent conjugation between DNA and the NLS peptide, rather than noncovalent or labile attachment strategies.

My comment regarding FACS analysis was considered through the use of transfected plasmids, capped linear gene cassettes, and peptide-modified gene cassettes that do not encode fluorescent proteins. These controls confirm that the gating strategy and experimental design are not biased by artifacts. That said, I would have appreciated the inclusion of double fluorescence gating.

The generalizability of the approach has been appropriately discussed. A more consolidated message involving primary or stem cells would have further strengthened the manuscript, although the authors provide a reasonable rationale for their current focus.

The remaining comments I previously raised have also been addressed.

Overall, the manuscript has been substantially improved, with the inclusion of additional Figures or supplementary data and an expanded discussion that reinforces the claims presented in the title.

At this stage, I have only minor comments remaining. These include clarifications such as specifying "nuclear copy number per cell" on the Y-axis of Figure 3d, and enhancing Figure quality and font size to improve readability.

Accordingly, I consider the manuscript suitable for publication in Nature Communications, pending minor revisions to address these final points.

Reviewer #3

(Remarks to the Author)

The manuscript is significantly improved with additional supporting evidence. Measurements of DNA copies in total and nuclear fractions, along with RNA transcript quantification, strengthen the validity of this new approach.

Minor editing issues:

In the text "the dual-end modification drastically reduced expression to levels below that of the SV40 KO control (<1%) (Figure 3d, Supplementary Fig. 5)", it is not appropriate to cite 3d here?

In Fig. 3d: it is better to directly label "Nuclear DNA copy number per cell" on the panel to avoid confusion.

Reviewer #1 (Remarks to the Author):

The manuscript of Mohamedshah et al describes a new method to deliver genes to cellular nuclei in order to induce expression of encoded proteins. Using their earlier developed method to replace a guanine with a 7-deazaguanine derivative, they changed it into an approach that incorporates the strained alkyne DBCO into the module. Subsequently, nuclear localization signal (NLS) peptide sequences are attached, after which the genes are introduced into the module. By means of various experiments they show how this approach allows the efficient delivery of a gene for the reporter system eGFP to the nucleus of human cells. Although much of the work focusses on HepG2 cells, two other cell lines were also used, illustrating the wider applicability of their approach. The approach was optimized along various axes, including linker length and type of NLS, and evidence is provided that a single mutation in the peptide sequence suppresses gene delivery, as was inferred from the absence of eGFP expression. Importantly, GFP-encoding genes were adequately delivered by this approach, indicating its suitability for the delivery of 7000 bp long genes.

All things considering it is a thoroughly performed study that has sufficient merit to be considered for publication in Nature Communications, provided the following points are addressed and the manuscript is properly revised.

[Authors reply] We appreciate the positive comments from the reviewer. We have carefully considered and addressed the concerns as detailed below.

1) A list of abbreviations needs to be included.

[Authors reply] We have now included a list of abbreviations that are not defined in the main text.

2) When the different set of NLS peptides is provided (e.g., in Figure 4a), please include a column with net charge of the peptide at pH 7.4 (using 1+ for H, K, R, and N-terminus; and 1- for D, E and C-terminus). While updating the table in Figure 4a, I suggest that the linker is displayed in a different font color (e.g., in grey). The authors should also mention why they did not simply acylate the N-terminus with an azide-containing group (which is the most direct approach), but followed a slightly less usual route using azidolysine at the C-terminus. There is probably a good reason for this, and the reader should be informed.

[Authors reply] We have updated the table in Figure 4a to now include the net charges of the peptides used along with changing the font color to grey for the linker portion.

Figure 4. Screening NLS peptides for enhanced gene expression in multiple cell types

(a) Suite of NLS peptide sequences selected for nuclear translocation of DNA cargos. Linker sequence ((GGGGS)₃K(N₃)) is colored grey. (b) Percent of growth-arrested HepG2, HEK293T, or AC16 cells with eGFP fluorescence 48 hours after transfection with peptide modified eGFP cassettes. Data collected in (b) were quantified using flow cytometry and are presented as mean ± s.d. for $n = 5$ biologically independent experiments, with individual data points shown, 100 ng DNA transfected per condition. Controls include eGFP plasmid DNA, unmodified, capped linear eGFP cassette, and SV40 KO modified eGFP cassette. All peptide modifications were on the 5' MluI side of the cassette with the 3x linker ((GGGGS)₃). Statistical analysis was performed using one-way ANOVA with Tukey's multiple comparison ($*p \leq 0.05$, $**p \leq 0.01$, $***p \leq 0.001$, $****p \leq 0.0001$, $ns p > 0.05$).

We chose to incorporate the azido group on the C-terminus of the peptide as the orientation for some of the NLS peptides does matter and the C-terminus is more amenable to modification (with both a flexible linker sequence and azido amino acid) while maintaining NLS activity. Furthermore, as we synthesized these peptides in-house, it was more efficient in terms of both yield and cost to incorporate an azido amino acid at the C-terminus. With the reviewer's comment in mind, we have now incorporated the following sentences into the manuscript:

(revised manuscript)

Peptides were synthesized by solid phase peptide synthesis including a flexible amino acid linker and azido amino acid at the C-terminus. These modifications to the NLS sequences were included at the C-terminus to minimize disruption of the NLS activity of the peptides while maintaining high yields and purity through synthesis.

3) At the end of the section entitled ‘Screening NLS peptides for enhanced gene expression’, the results are tentatively connected to importin-alpha isoforms. What are the expression levels of these isoforms in the different cell lines that were used? Also, it is very likely that also linker-length (and composition!) affect transfection in different cell types. Therefore, I encourage the authors to expand this discussion with a few sentences to at least mention all variables that might be at play.

[Authors reply] We thank the reviewer for this insightful comment regarding importin- α isoform expression with regards to NLS mediated nuclear translocation of DNA cargos. Transcriptomic data indicate that importin- α isoforms (KPNA1-7) exhibit cell- and tissue-specific expression patterns, although detailed profiles for all individual cell types remain uncharacterized. For example, KPNA1, KPNA2, and KPNA 6 are well expressed in HepG2s, though similar data is not as readily available for AC16s or HEK293Ts. Further, the binding mechanisms of the importin- α isoforms to various NLS peptides is poorly understood, making it difficult to predict which NLS peptides may be preferred by specific cell- or tissue-types based on importin- α isoform expression. Variations in importin- α type and abundance may influence the binding and transport of individual DNA-NLS conjugates, in conjunction with the chemical composition and linker design of the NLS peptide itself. With this in mind, we have expanded upon the discussion in the revised manuscript to include these considerations.

(original manuscript)

Furthermore, expression differences across cell types may stem from variations in endogenous nuclear transport factors, particularly importin- α isoforms.^{66–68} The human genome encodes seven distinct importin- α isoforms with tissue-specific expression, which may explain the observed variance.^{67–70} Different NLS peptides likely exhibit varying affinities for these isoforms, though their exact binding mechanisms remain unexplored.

(revised manuscript)

The observed cell type-specific variability in eGFP expression may also arise from differences in expression profiles of endogenous importin- α isoforms (KPNA1-7), which are known to vary across cell and tissue types including hepatic, renal, and cardiac tissues.^{70–74} For example, *KPNA1*, *KPNA2*, *KPNA6* are known to be expressed in HepG2s, whereas *KNPA7* is minimally expressed.^{75,76} Such variation in importin- α isoform type and abundance could influence the binding affinities and, consequently, the transport efficiencies of individual NLS peptides

conjugated to DNA cargos. However, the specific binding mechanisms between different NLS peptides and importin- α remain under-studied, making it difficult to predict NLS preference for a specific cell or tissue type given importin- α isoform expression levels. Additionally, factors such as peptide composition and linker length may further modulate NLS accessibility and nuclear import efficiency in a cell-specific manner that depends on importin- α isoform expression, as both peptide flexibility and local charge environments can impact recognition by importin complexes.

4) The observation that the dual-end modified constructs lead to expression levels below the Sv40 KO negative control is very intriguing. If it is indeed the case that the peptides engage two different nuclear pore complexes, essentially glueing it to the outside of the nuclear envelope, this effect might be countered by including shorter linker, i.e., a single GGGGS linker. It would be interesting to see the result of such an experiment, assuming this can be performed conveniently.

[Authors reply] We thank the reviewer for this insightful comment to better understand the impact of dual-end NLS modified gene cassettes. We repeated this experiment with dual-end NLS modified gene cassettes in which the linker length was increased or decreased relative to the Lx3 linker. We observed no differences in eGFP expression levels for these new constructs compared to the Lx3 data originally presented in Figure 3c. We have now referred to this data in the manuscript and updated the Supplementary Data to now include the following figure (Supplementary Fig. 5).

Supplementary Figure 5. Screening eGFP expression after 48 h in growth arrested HepG2 cells with dual end modified NLS peptides of varying linker length (GGGG(S)_n).

Data collected were quantified using flow cytometry and are presented as mean \pm s.d. for $n = 5$ biologically independent experiments, individual data points are overlaid, 100 ng DNA transfected per condition. Statistical analysis was performed using one-way ANOVA with Tukey's multiple comparison ($*p \leq 0.05$, $**p \leq 0.01$, $***p \leq 0.001$, $****p \leq 0.0001$, ns $p > 0.05$).

5) Interestingly, the authors find that constructs that contain two NLS peptides do not lead to efficient eGFP production. This directly relates to a comment in the introduction of multiple NLS peptides per DNA cargo, which states that 'non-site-specific modification that can lead to multiple NLS peptides per DNA cargo.', and I encourage the authors to relate to this when this is treated in the text (e.g., just above the sub-heading 'Screening NLS peptides for enhanced gene expression').

[Authors reply] We thank the author for this comment. We have expanded on the discussion of multiple NLS peptides per DNA cargo and site-specificity of the NLS modification on DNA. We have also incorporated a new experiment involving electrostatic conjugates of NLS peptides and DNA as suggested by Reviewer 2.

(revised manuscript)

To benchmark our system, we adapted the strategy of a previous study by incubating our gene cassettes with NLS peptides (allowing them to electrostatically interact) and assessed eGFP expression following lipofection-mediated transfection into growth-arrested HepG2 cells.³⁶ Electrostatic interactions allow for multiple NLS peptides to interact with DNA, albeit with little control of the number or location of peptides on a DNA copy. No significant differences in expression ($p > 0.05$) were observed between electrostatically associated NLS-DNA cassettes and controls (Supplementary Fig. 6). Together with the lack of enhancement observed for dual NLS-modified gene cassettes, these results indicate that a single, site-specific NLS modification is optimal for promoting gene expression through improved nuclear translocation.

Supplementary Figure 6. Screening eGFP expression of electrostatically conjugated NLS peptides to DNA gene cassettes after 48 h in growth arrested HepG2 cells.

Data collected were quantified using flow cytometry and are presented as mean \pm s.d. for $n = 5$ biologically independent experiments, individual data points are overlaid, 100 ng DNA transfected per condition. Statistical analysis was performed using one-way ANOVA with Tukey's multiple comparison ($*p \leq 0.05$, $**p \leq 0.01$, $***p \leq 0.001$, $****p \leq 0.0001$, ns $p > 0.05$).

6) In their design, the authors also screened various linker lengths. They observe that shorter linkers lead to less efficient eGFP expression than longer linkers, and they attribute it to a better charge separation. How does an increased linker length lead to a better charge separation? One could also argue that a longer spacer leads to more flexibility in the connection between peptide and DNA, which might actually facilitate charge interactions.

[Authors reply] We thank the reviewer for this insightful comment. We initially hypothesized that the enhanced expression observed with longer linkers arose primarily from increased charge separation between the negatively charged DNA and the positively charged NLS peptide. Upon further consideration, we believe that the greater physical distance and flexibility provided by the longer linker likely improves the accessibility of the NLS peptide to nuclear transport factors, with charge separation contributing secondarily to this enhanced accessibility. We have shortened the discussion discussing charge separation and have included the following into the manuscript:

(original manuscript)

Specifically, as many NLS peptides, particularly classical NLS peptides, are positively charged, it is thought that the positively charged peptide electrostatically interacts with negatively charged DNA, potentially preventing the NLS peptide from binding with appropriate nuclear translocation factors and reducing the effectiveness of nuclear translocation.

This reinforces the role of charge separation in DNA-NLS nuclear translocation

(revised manuscript)

These results suggest that increasing the flexible linker length, up to an optimal point, enhances DNA-NLS nuclear translocation by improving the accessibility of the NLS peptide to nuclear transport factors, with the increased physical distance, flexibility, and potential charge separation between the DNA and peptide likely contributing to this effect.

7) The paper solely relies on Urea-PAGE for purity analysis of the obtained constructs. This type of analysis is heavily biased to the visualization of DNA (GelRed is a nucleic acid staining), and neglects the peptides. Although I have no reason to doubt that the proper constructs have been formed, I strongly encourage the authors to include several reversed-phase HPLC chromatograms of the prepared DNA-NLS conjugates, i.e., at least of the most-used constructs and the mutated negative control.

[Authors reply] We thank the reviewer for this insightful comment regarding the characterization of the DNA-NLS conjugates. Following the click reaction, the conjugates were purified using standard DNA cleanup columns to remove residual salts prior to analysis. We attempted to analyze DNA-peptide conjugates using reverse-phase HPLC; however, these methods could not be validated due to the lack of access to appropriate DNA-compatible columns. Urea-PAGE analysis showed minimal impurities, and the purified DNA-NLS conjugates were directly injected for high-resolution mass spectrometry (HRMS), which confirmed the expected molecular weights with minimal detectable contaminants (Supplementary Information). Based on these results, we have high confidence in the purity of the generated DNA-NLS conjugates.

8) The authors mention that the DNA-NLS conjugates were obtained in 90-95% yield. However, this is based on DNA, and not on the peptide (the latter is used in 5-fold excess). Also, this is based on the assumption the staining of the DNA-NLS conjugates is not affected by the presence of the peptide, whereas this is not necessarily the case. Evidence that GelRed staining works equally well on DNA and on DNA-NLS conjugates should be provided (could also be a literature reference to a dedicated study on this aspect) before this yield percentage can be reported. Alternatively, another analytical technique needs to be applied for quantitative analysis.

[Authors reply] We thank the reviewer for this helpful comment regarding the yield determination of the DNA-NLS conjugates. The near-quantitative yields typically obtained from strain-promoted azide-alkyne cycloaddition reactions are well established, and we used a five-fold excess of peptide to further ensure high conjugation efficiency. While relevant literature describing GelRed staining of DNA-peptide conjugates is limited, we have previously validated the use of GelRed band-intensity comparison to estimate yields of TGT-modified DNA stem loops (Tota, E. M. & Devaraj, N. K. *J. Am. Chem. Soc.* **145**, 8099–8106, 2023).

To directly assess whether peptide attachment affects GelRed staining, we re-performed Urea-PAGE analysis comparing an unmodified DNA stem loop, a DNA-peptide conjugate, DNA mixed with peptide (without conjugation), and peptide alone. Gels were stained with both GelRed and SYBR Safe (Supplementary Fig. 2). No significant differences in band intensity were observed with either dye, indicating that peptide conjugation does not alter staining efficiency. These results support that the DNA-NLS conjugates were generated with high purity and near-quantitative yield.

Supplementary Figure 2. Urea-PAGE of DNA oligonucleotide – peptide conjugates with different staining dyes.

Urea-PAGE analysis of TGT-mediated labeling of MluI 170 PP DNA oligonucleotide with preQ₁-DBCO and peptides. Upward gel shift indicates successful covalent labeling. (A) depicts a gel stained with GelRed. Lane 1 – MluI 170 PP-DBCO, Lane 2 – MluI 170 PP-extSV40, Lane 3 – MluI 170 PP + extSV40 (not conjugated), and Lane 4 – extSV40. (B) depicts a gel stained with SYBER Safe. Lane 1 – MluI 170 PP-DBCO, Lane 2 – MluI 170 PP-extSV40, Lane 3 – MluI 170 PP + extSV40 (not conjugated), Lane 4 – extSV40, Lane 5 – MluI 170 PP-DBCO, Lane 6 – MluI 170 PP-PLSCR-1, Lane 7 – MluI 170 PP + PLSCR-1 (not conjugated), and Lane 8 – PLSCR-1.

9) Why do the ESI-TOFMS spectra of the Sv40 NLS peptides with linkers (G4S)_n (where n>1) show peaks corresponding to shorter lengths? Specifically Sv40 NLS Lx2 shows peaks from Sv40 NLS Lx1 and Sv40 NLS Lx3 shows peaks of both Sv40 NLS Lx2 and of Sv40 NLS Lx1. Along similar lines, the spectrum for Sv40 NLS Lx4 shows a peak assigned as [M+H+2Na]³⁺ at 972.7300, which could equally originate from Sv40 NLS Lx1 [M+2H]²⁺ (which appears at 973.0276 in the first spectrum and tentatively at 972.5192 in the third spectrum). Similar observations can be made for ESI-TOFMS spectra of most other NLS-peptides in which linker-lengths are varied. I assume these result from anomalies in the analysis, and do not originate from the samples themselves (i.e., the applied conjugates are unique and clean, and do not contain a mixture of components). If, however, evidence of this cannot be provided, it is hard to see how the presented results originate from the indicated conjugates.

[Authors reply] We thank the reviewer for their careful observation regarding the discrepancies in the ESI-TOFMS spectra. The samples have been reanalyzed, and the updated Supplementary Data now include the corrected spectra. We believe the previously observed overlap between spectra of similar peptides resulted from sample carry-over during sequential runs. In the reanalysis, special care was taken to prevent this issue by running solvent blanks between each sample to eliminate carry-over and ensure accurate mass assignments.

10) Change Fmoc-AA-OH into Fmoc-AA(PG)-OH to include the protecting group that is also needed for several amino acids.

[Authors reply] We thank the reviewer for this comment and have made the recommended changes.

Additional comments on the figures.

Figure 1, panel a: three different strategies are shown, but they are not clearly separated. Please add visual aids to separate the approaches. The text at the bottom describes that the DNA-NLS conjugate is expressed, but that is not the case. The gene that is carried inside by the DNA-NLS conjugate is expressed. Please rephrase to increase accuracy. Also, the arrow that shows passage of the NPC can be made bolder (also in panel b).

[Authors reply] Thank you for this comment to help clarify Figure 1. We have made the suggested modifications to improve clarity and accuracy.

Figure 1. An overview of DNA-NLS conjugates for nuclear translocation of DNA cargos.

(a) Summary of DNA-NLS conjugate strategies previously explored for nuclear translocation of a DNA cargo through the nuclear pore complex (NPC). DNA-NLS conjugates are summarized in three broad categories: NLS-tagged nuclear proteins associated electrostatically with DNA cargo, noncovalent interactions between DNA cargo and NLS peptides, and covalent interactions between DNA cargo and NLS peptides. (b) Overview of NLS-modified gene cassettes generated by DNA-PepTAG. DNA-PepTAG conjugates rely on a nonreversible, biocompatible triazole linkage between the NLS peptide and DNA cargo of interest.

Figure 2, caption panel b: the right part of panel b does not show ‘their covalent attachment to a DNA stem loop by TGT’, but ‘PAGE-urea gel analysis of the covalent attachment to a DNA stem loop by TGT’. Please rephrase, and rephrase ‘Urea-PAGE analysis shows an’ in the last sentence of this part with ‘The’.

[Authors reply] We thank the reviewer for their comment to improve the clarity for the caption of Figure 2. We have made the following changes:

(original manuscript)

(b) Chemical structures of preQ₁ probes (left) and their covalent attachment to a DNA stem loop by TGT (right). PreQ₁-biotin serves as a positive control for DNA-TAG activity. Urea-PAGE analysis shows an upward gel shift between negative control (MluI 170 PP DNA hairpin, lane 1) and preQ₁-biotin (lane 2) and preQ₁-DBCO (lane 3) with near quantitative yields.

(revised manuscript)

(b) Chemical structures of preQ₁ probes (left) and their covalent attachment to a DNA stem loop via TGT by Urea-PAGE (right). PreQ₁-biotin serves as a positive control for DNA-TAG activity. The upward gel shift between the negative control (MluI 170 PP DNA hairpin, lane 1) and preQ₁-biotin (lane 2) and preQ₁-DBCO (lane 3) suggesting near quantitative yields.

Figure 3. The list of features in panel a follows a different order than given in the caption. Please align. Also, a comment should be provided on the origin of the background in Fig3b negative controls?

[Authors reply] We thank the reviewer for their comment to improve the clarity of the caption for Figure 3. We have made the following changes:

(original manuscript)

(a) Parameters of NLS-modified eGFP gene cassettes were optimized: (GGGS)_n linker length (L_n), peptide sequences, and single/dual(MIuI/AvrII) conjugation. Linear eGFP cassettes were

transfected into aphidicolin growth arrested HepG2 cells, and eGFP expression was monitored by microscopy and flow cytometry. (b) Representative epifluorescence microscopy images showing eGFP expression in growth-arrested HepG2 cells 48 hours after transfection with linearized eGFP cassette. Shown are capped, linearized cassettes with no peptide (linear control), an Sv40 KO peptide (a single amino acid substitution, K→T, from the Sv40 NLS that is known to inhibit nuclear translocation), and three example NLS peptides.

(revised manuscript)

(a) Parameters of NLS-modified eGFP gene cassettes were optimized: (GGGGS)_n linker length (L_n), single/dual(MIuI/AvrII) conjugation, and peptide sequences. Linear eGFP cassettes were transfected into aphidicolin growth arrested HepG2 cells, and eGFP expression was monitored by microscopy and flow cytometry. (b) Representative epifluorescence microscopy images showing eGFP expression in growth-arrested HepG2 cells 48 hours after transfection with linearized eGFP cassette. Shown are capped, linearized cassettes with no peptide (linear control), an SV40 KO peptide (a single amino acid substitution, K→T, from the SV40 NLS that is known to inhibit nuclear translocation), and three example NLS peptides. “Linear control” is the capped, unmodified linear eGFP cassette.

Figure 4. Statistical analysis is provided on the bars with the largest differences. I think this should also be done between bars with the smallest differences, so that the reader understands the range of differences.

[Authors reply] We thank the reviewer for this comment regarding the statistical analyses presented in the figure. To maintain visual clarity, we have omitted statistical annotations for less relevant comparisons, as including all would overly clutter the graphs. A complete statistical analysis, including all individual comparisons and corresponding *p*-values, will be provided in the final version of the manuscript in accordance with *Nature Communications* data presentation guidelines.

Figure 5. The end of the caption refers to details on how data for the results in panel (c) was obtained. This should move to the caption directly associated to panel (c), and not placed at panel (d).

[Authors reply] We thank the reviewer for this comment. The following changes have been made:

(original manuscript)

(c) Concentrations of secreted factor IX protein in serum from growth-arrested HepG2 cells 48 hours after transfection with DNA cassettes encoding factor IX. Controls included untransfected cells (labeled as “negative control”), factor IX-encoding plasmid, unmodified, capped linear cassette, and Sv40 KO modified cassette. (d) Secretion of factor IX from growth-arrested HepG2 cells at time points up to 120 hours following transfection of NLS-modified cassettes. Controls include untransfected cells (labeled as “negative control”), unmodified, capped linear cassette, and

Sv40 KO modified cassette. Data collected in (c) and were obtained by ELISA and presented as a mean \pm s.d. for $n = 4$ biologically independent experiments, with individual data points shown, 100 ng DNA transfected per condition. Statistical analysis was performed using one-way ANOVA with Tukey's multiple comparison ($*p \leq 0.05$, $**p \leq 0.01$, $***p \leq 0.001$, $****p \leq 0.0001$, ns $p > 0.05$). Data collected in (d) and were obtained by ELISA and presented as a mean \pm s.d. for $n = 3$ biologically independent experiments, 100 ng DNA transfected per condition.

(revised manuscript)

(c) Concentrations of secreted factor IX protein in serum from growth-arrested HepG2 cells 48 hours after transfection with DNA cassettes encoding factor IX. Controls included untransfected cells (labeled as "negative control"), factor IX-encoding plasmid, unmodified, capped linear cassette, and SV40 KO modified cassette. Data collected in (c) and were obtained by ELISA and presented as a mean \pm s.d. for $n = 4$ biologically independent experiments, with individual data points shown, 100 ng DNA transfected per condition. Statistical analysis in (a), (b), and (c) was performed using one-way ANOVA with Tukey's multiple comparison ($*p \leq 0.05$, $**p \leq 0.01$, $***p \leq 0.001$, $****p \leq 0.0001$, ns $p > 0.05$). (d) Secretion of factor IX from growth-arrested HepG2 cells at time points up to 120 hours following transfection of NLS-modified cassettes. Controls include untransfected cells (labeled as "negative control"), unmodified, capped linear cassette, and SV40 KO modified cassette. Data collected in (d) and were obtained by ELISA and presented as a mean \pm s.d. for $n = 3$ biologically independent experiments, 100 ng DNA transfected per condition.

Reviewer #2 (Remarks to the Author):

MAJOR REVISION RECOMMENDED

In their manuscript entitled "Highly Efficient Expression of DNA-peptide Conjugates in Growth-arrested Cells", Zulfiqar Y. Mohamedshah et al. present a procedure for efficiently obtaining DNA-peptide assemblies designed to overcome the nuclear membrane barrier, which is achieved through nuclear localization signal (NLS) mediating active nuclear import in non-dividing cells. According to the authors, the proposed workflow underscores the potential of DNA-NLS conjugates for non-viral gene therapy applications.

While this study appears both innovative and engaging, I have several comments and questions – primarily concerning the experimental aspects – that require authors' attention. To better convince the reader and enhance the study impact, I think additional investigations and analyses are necessary. Consequently, I recommend a major revision to address the comments outlined below.

[Authors reply] We appreciate the positive comments from the reviewer. We have carefully considered and addressed the concerns as detailed below. We believe the included experiments have significantly strengthened the manuscript as suggested by the reviewer.

Main Comments:

A key message of this paper is “enabling enhanced nuclear delivery of therapeutic genes while minimizing the required DNA dose”. The potential of the approach is demonstrated through the expression of eGFP or Factor IX cassettes carried by the delivered DNA cargos, both serving as reporter systems i.e., systems reporting the efficiency of the trafficking into the nucleus. While the results offer valuable insights, I think they do not provide sufficient direct evidence of enhanced nuclear import. A more definitive demonstration could be achieved by comparing the copy number of DNA-peptide conjugates in the nucleus with that in the cytoplasm, across different experimental conditions. This could involve subcellular fractionation, quantitative PCR, immunofluorescence, or other relevant methods.

[Authors reply] We thank the reviewer for this insightful comment emphasizing the need for direct evidence of increased nuclear translocation of NLS-modified gene cassettes. To address this, we transfected NLS-modified gene cassettes and relevant controls (plasmid, linear control, and SV40 KO modified cassette) into growth-arrested HepG2 cells. 48 h post-lipofection, cells were gently lysed, and nuclear and extranuclear fractions were isolated. Quantitative PCR was then performed to determine the copy number of transfected DNA in each fraction. We observed a significant ($p < 0.05$) increase (~10–20 fold) in nuclear DNA copies in cells transfected with NLS-modified cassettes compared to controls. This provides strong evidence that NLS modification enhances nuclear import of DNA cargo, consistent with the elevated expression levels previously reported.

Additionally, total transfected DNA and mRNA transcripts were quantified by qPCR across different gene cassettes, as suggested by Reviewer 3, further supporting that NLS conjugation improves nuclear translocation and subsequent transcription.

Based on these findings, we have revised Figure 3 to include the nuclear DNA copy number data (now presented as Fig. 3d) and expanded the discussion accordingly.

Figure 3. Optimization and Nuclear Localization of NLS-modified gene cassettes

(a) Parameters of NLS-modified eGFP gene cassettes were optimized: $(GGGGS)_n$ linker length (L_n), single/dual(MIuI/AvrII) conjugation, and peptide sequences. Linear eGFP cassettes were transfected into aphidicolin growth arrested HepG2 cells, and eGFP expression was monitored by microscopy and flow cytometry. (b) Representative epifluorescence microscopy images showing eGFP expression in growth-arrested HepG2 cells 48 hours after transfection with linearized eGFP

cassette. Shown are capped, linearized cassettes with no peptide (linear control), an SV40 KO peptide (a single amino acid substitution, K→T, from the SV40 NLS that is known to inhibit nuclear translocation), and three example NLS peptides. “Linear control” is the capped, unmodified linear eGFP cassette. Scale bar: 50 μm. (c) Percent of growth-arrested HepG2 cells with eGFP fluorescence 48 hours after transfection with NLS-modified DNA cassettes in which the NLS (SV40 or Influenza) is on the 5' MluI side (peptide A), 3' AvrII side (peptide B), or both sides (peptides A and B). The “SV40 KO” negative control is the linear cassette modified on the 5' MluI side with the SV40 KO peptide. All peptides utilize the 3x linker ((GGGGS)₃). (d) Copy number of the eGFP DNA gene cassette in the nucleus (per cell) 48 hours after lipofection. Peptide modifications are on the 5' MluI side of the gene cassette with the 3x linker ((GGGGS)₃). Controls include eGFP plasmid, unmodified, capped linear cassette, and the SV40 KO modified cassette. (e) Percent of growth-arrested HepG2 cells with eGFP fluorescence 24 hours after transfection with eGFP cassettes with or without 10 μM importazole treatment. Peptide modifications are on the 5' MluI side of the gene cassette with the 3x linker ((GGGGS)₃). Controls include eGFP plasmid, unmodified, capped linear cassette, and the SV40 KO modified cassette. Data collected in (c) and (e) were quantified using flow cytometry and are presented as mean ± s.d. for $n=5$ biologically independent experiments, with individual data points shown, 100 ng DNA transfected per condition. Data collected in (d) were quantified using qPCR, normalized against RPP30 housekeeping gene, and are presented as mean ± s.d. for $n=3$ biologically independent experiments, with individual data points shown, 100 ng DNA transfected per condition. For (c) and (d), statistical analysis was performed using one-way ANOVA with Tukey's multiple comparison ($*p \leq 0.05$, $**p \leq 0.01$, $***p \leq 0.001$, $****p \leq 0.0001$, $ns p > 0.05$). For (e), statistical analysis was performed using multiple unpaired t -tests ($*p \leq 0.05$, $**p \leq 0.01$, $***p \leq 0.001$, $****p \leq 0.0001$, $ns p > 0.05$).

(revised manuscript)

We quantified the copy number of exogenous DNA translocated to the nucleus and found that NLS-modified gene cassettes had significantly enhanced nuclear delivery compared to controls.

While these results strongly suggest that NLS-modified DNA cassettes enhance expression by promoting nuclear translocation, they do not directly demonstrate increased nuclear import. To quantify total DNA delivery to growth-arrested HepG2 cells, we lysed cells 48 hours post-transfection and performed quantitative PCR (qPCR) to determine absolute DNA copy numbers. There was no significant difference ($p > 0.05$) in total intracellular DNA between NLS-modified and unmodified cassettes, indicating that enhanced gene expression from NLS-modified constructs is not simply due to increased cellular uptake via lipofection (Supplementary Figs. 7, 8). However, qPCR analysis of isolated nuclear fractions revealed a significant ($p < 0.05$) enrichment (~10-20 fold) of NLS-modified DNA compared to controls (Fig. 3d), providing direct evidence that NLS conjugation facilitates nuclear import of DNA cargo. The observed increase in nuclear copy number, from hundreds to thousands of DNA molecules per nucleus, aligns with previously reported thresholds required for robust gene expression.^{20,57-60} This correlation is in agreement with the elevated gene expression we observed for NLS-modified gene cassettes. Consistent with this, eGFP mRNA quantification from transfected cells confirmed that enhanced nuclear accumulation of NLS-modified DNA cassettes corresponds to both higher transcription and subsequent expression (Supplementary Fig. 9).

Additionally, the following Supplementary Figures were included:

Supplementary Figure 7. Total eGFP DNA cassette copy number delivered to growth-arrested HepG2 cells 48 h following lipofection.

Data collected were quantified using qPCR, normalized against RPP30 as a housekeeping gene, and are presented as mean \pm s.d. for $n = 5$ biologically independent experiments, with individual data points shown, 100 ng DNA transfected per condition. Statistical analysis was performed using one-way ANOVA with Tukey's multiple comparison ($*p \leq 0.05$, $**p \leq 0.01$, $***p \leq 0.001$, $****p \leq 0.0001$, ns $p > 0.05$).

Supplementary Figure 8. Extranuclear, nuclear, and total eGFP DNA gene cassette copy numbers delivered to growth-arrested HepG2 cells 48 h following lipofection. The nuclear fraction of lysate was separated from the extranuclear fraction via centrifugation (see methods).

Data collected were quantified using qPCR, normalized against RPP30 as a housekeeping gene, and are presented as mean \pm s.d. for $n = 5$ biologically independent experiments, with individual data points shown, 100 ng DNA transfected per condition. Nuclear DNA copies are also presented in Figure 3d. Total delivered DNA copies is a summation of extranuclear and nuclear DNA copies. Statistical analysis was performed using one-way ANOVA with Tukey's multiple comparison ($*p \leq 0.05$, $**p \leq 0.01$, $***p \leq 0.001$, $****p \leq 0.0001$, $ns p > 0.05$).

Supplementary Figure 9. mRNA transcripts copy numbers from eGFP DNA gene cassettes delivered to growth-arrested HepG2s via lipofection.

Data collected were quantified using qPCR, normalized against GAPDH as a housekeeping gene, and are presented as mean \pm s.d. for $n = 5$ biologically independent experiments, with individual data points shown, 100 ng DNA transfected per condition. Statistical analysis was performed using one-way ANOVA with Tukey's multiple comparison ($*p \leq 0.05$, $**p \leq 0.01$, $***p \leq 0.001$, $****p \leq 0.0001$, $ns p > 0.05$).

We believe the inclusion of these experiments and the data generated provides strong evidence supporting that our NLS-modifications on gene cassettes facilitate greater nuclear import of a DNA cargo.

The authors state “the use of growth-arrested cells allows direct assessment of the efficacy of NLS peptides in facilitating nuclear entry”. Growth arrest was achieved by inhibiting cell division with aphidicolin. For certain assays, importazole was additionally used to inhibit importin- β , thereby blocking nuclear import. Additionally, it is mentioned in SI “Expression was assessed only at 24 h and not longer due to the cytotoxic properties of importazole.”. This raises questions regarding these drugs and their possible direct or indirect effects on the reported results. Exploring other cell division inhibitors, such as nocodazole or colchicine, and comparing results with non-growth-arrested cells could provide complementary valuable insights.

[Authors reply] We thank the reviewer for this insightful comment regarding the choice of growth-arrest agent and the comparison to actively dividing cells. To address this, we evaluated eGFP expression from NLS-modified gene cassettes in both nocodazole-arrested and actively dividing HepG2 cells (Supplementary Fig. 10). Expression profiles were similar between nocodazole and aphidicolin growth-arrested cells, indicating that the specific growth arrest agent does not affect nuclear translocation of DNA cargo. As expected, transfection of actively dividing cells yielded comparable expression levels across all constructs, reinforcing that growth arrest is essential for isolating and assessing NLS-mediated nuclear import. These new data and the corresponding discussion have been included in the revised manuscript (Supplementary Fig. 10).

(revised manuscript)

Finally, in the experiments presented here, growth arrest was achieved using aphidicolin, which inhibits DNA polymerase and arrests the cell cycle at the G1/S phase. To assess whether the specific method of growth arrest influences nuclear translocation of DNA cargo, we compared eGFP expression from NLS-modified gene cassettes in HepG2 cells treated with nocodazole, which arrests cells in M phase. Expression levels in nocodazole-treated cells were comparable to those observed in aphidicolin-arrested HepG2 cells (Fig. 4b, Supplementary Fig. 10). In contrast, actively dividing HepG2 cells showed no significant differences ($p > 0.05$) in eGFP expression between NLS-modified gene cassettes and controls (Supplementary Fig. 10). These findings underscore the importance of arresting the cell cycle in isolating NLS-mediated nuclear import, as nuclear envelope breakdown during mitosis can otherwise facilitate passive entry of DNA cargo into the nucleus.^{17–20}

Additionally, the following supplementary figure was included:

Supplementary Figure 10. Screening eGFP expression after 48 h in nocodazole growth arrested or actively dividing HepG2 cells with NLS-modified gene cassettes.

(A) Percent of nocodazole growth-arrested HepG2s with eGFP fluorescence 48 hours after transfection. (B) Percent of actively dividing HepG2s with eGFP fluorescence 48 hours after transfection. Data collected were quantified using flow cytometry and are presented as mean \pm s.d. for $n = 5$ biologically independent experiments, individual data points are overlaid, 100 ng DNA

transfected per condition. Statistical analysis was performed using one-way ANOVA with Tukey's multiple comparison (* $p \leq 0.05$, ** $p \leq 0.01$, *** $p \leq 0.001$, **** $p \leq 0.0001$, ns $p > 0.05$).

Figure 1 presents a comparison of DNA-NLS conjugates, emphasizing the limitations of previous systems. To better appreciate the results reported and further substantiate the claimed superiority of the approach presented in this study, it would have been valuable to include comparative results of some previously reported DNA-NLS conjugates tested in parallel.

[Authors reply] We thank the reviewer for this comment regarding benchmarking our DNA-NLS conjugates against previously reported systems. We did not benchmark directly against the most widely cited covalent conjugation approach utilizing a maleimide linkage between the NLS and DNA (Zanta *et al.*, *Proc. Natl. Acad. Sci. USA* **96**, 91–96, 1999), as this strategy has since been replicated with no observed enhancement in nuclear entry of DNA-NLS conjugates (Tanimoto *et al.*, *Bioconjugate Chem.* **14**, 1197–1202, 2003).

Instead, we sought to compare our DNA-NLS conjugates with those generated through non-specific electrostatic interactions, which were more feasible to reproduce experimentally. Following the methodology described by Subramanian *et al.* (*Nat. Biotechnol.* **17**, 873–877, 1999), we electrostatically complexed NLS peptides, including several of our positively charged peptide variants as well as the best-performing PLSCR-1 NLS, to the linear eGFP cassette. Upon lipofection into growth-arrested HepG2 cells, we observed no significant differences ($p > 0.05$) in eGFP expression between electrostatically conjugated NLS–DNA cassettes and controls.

These results indicate that enhanced gene expression via nuclear translocation requires a single, site-specific, and stable covalent conjugation between DNA and the NLS peptide, rather than non-covalent or labile attachment strategies. We have included new discussion and a Supplementary Figure in the revised manuscript, integrating these results into the broader context of multiple NLS peptide modifications on DNA cargos.

(revised manuscript)

To benchmark our system, we adapted the strategy of a previous study by incubating our gene cassettes with NLS peptides (allowing them to electrostatically interact) and assessed eGFP expression following lipofection-mediated transfection into growth-arrested HepG2 cells.³⁶ Electrostatic interactions allow for multiple NLS peptides to interact with DNA, albeit with little control of the number or location of peptides on a DNA copy. No significant differences in expression ($p > 0.05$) were observed between electrostatically associated NLS-DNA cassettes and controls (Supplementary Fig. 6). Together with the lack of enhancement observed for dual NLS-modified gene cassettes, these results indicate that a single, site-specific NLS modification is optimal for promoting gene expression through improved nuclear translocation.

Supplementary Figure 6. Screening eGFP expression of electrostatically conjugated NLS peptides to DNA gene cassettes after 48 h in growth arrested HepG2 cells.

Data collected were quantified using flow cytometry and are presented as mean \pm s.d. for $n = 5$ biologically independent experiments, individual data points are overlaid, 100 ng DNA transfected per condition. Statistical analysis was performed using one-way ANOVA with Tukey's multiple comparison ($*p \leq 0.05$, $**p \leq 0.01$, $***p \leq 0.001$, $****p \leq 0.0001$, ns $p > 0.05$).

The authors draw conclusions and claim the “generability” of their approach on the basis of results obtained from three human cell lines. I wonder about its broader applicability, particularly in primary cells, which could offer further valuable insights into nuclear import efficiency under more physiologically relevant conditions.

[Authors reply] We thank the reviewer for this insightful comment regarding the generalizability of our approach across different cell types, particularly primary cells. Unfortunately, we did not have ready access to primary cells for this study. While we agree that assessing DNA-NLS conjugate performance in additional and more physiologically relevant cell types would provide valuable insights, we view this work primarily as a proof of principle study. Our focus here was to establish a robust and generalizable methodology for generating DNA-peptide conjugates via the DNA-PepTAG platform and to demonstrate that NLS modifications can enhance nuclear translocation and gene expression of DNA cargos. Expanding this approach to include a broader range of cell types, including primary and stem cells, represents an exciting direction for future studies.

In most experiments, the transfection efficiency of the various DNA-peptide conjugates was assessed by FACS analysis using an eGFP reporter system, where its expression served as a proxy for successful transfection and readout to monitor differences. While the percentage of positively transfected cells can indeed provide a measure of transfection efficiency, particular caution is warranted when using GFP as a marker. Considering Supplementary Figures 4–6, clarification of the gating strategy used to distinguish between GFP-negative and GFP-positive cells is needed. A dual-gate fluorescence analysis (FL1-H vs. FL2-H) is recommended to account for the inherent autofluorescence of cells and its potential variations, especially following transfection with cationic lipids. Such an approach would help exclude contributions from artifacts, for instance alterations in natural endogenous autofluorescence induced by the transfection reagent itself (see

for examples, DOI: 10.2144/01312st02 and DOI: 10.1006/abio.2000.5006). Additionally, incorporating an alternative reporter system such as luciferase would help consolidate and validate the findings further.

[Authors reply] We thank the reviewer for this thoughtful comment regarding potential autofluorescence during flow cytometry analysis. We are confident that our gating strategy effectively excludes autofluorescent events arising from either cell death or transfection related artifacts. To confirm this, we additionally transfected plasmids, capped linear gene cassettes, and peptide-modified gene cassettes that do not encode fluorescent proteins to assess any background eGFP signal. As shown in the representative flow plots below, we observed negligible eGFP-positive populations in these negative controls, indicating that our gating strategy and control experiments successfully eliminate false-positive fluorescence and accurately distinguish true eGFP-expressing cells.

Further, we have updated the figure captions for Supplementary Figures 11-13 to clarify the gating strategy used to differentiate between eGFP negative and positive cells.

(revised manuscript)

The negative control was used to set the thresholding gate for eGFP positive cells as untransfected cells would have minimal fluorescence following elimination of cell debris which may be autofluorescent.

Other comments:

The abstract mentions non-viral gene therapy; however, the transfection reagent used, i.e., Lipofectamine 3000, is referenced only once in the manuscript, in the Methods section (page 22). To ensure clarity and coherence, its role should be highlighted more prominently, starting from the abstract. Additionally, the study could be completed with some characterizations of the lipoplexes formed by Lipofectamine 3000 and the various DNA-peptide conjugates. Parameters such as size and charge, as well as results of DNA retardation assays, could be incorporated to provide a more comprehensive analysis.

[Authors reply] We thank the reviewer for this helpful comment regarding the use of Lipofectamine 3000. We have now revised the manuscript to more prominently reference Lipofectamine 3000 and lipofection throughout the Abstract, Introduction, Results and Discussion, and Conclusion sections.

While additional characterization of the lipoplexes formed between DNA-peptide conjugates and Lipofectamine 3000 (e.g., size, charge, and DNA retardation assays) would indeed provide further insight, such analyses are beyond the scope of this study. We view the present work as a proof of principle demonstration of the DNA-PepTAG platform for generating DNA-NLS conjugates and establishing that these constructs enhance gene expression through increased nuclear translocation. Future studies will focus on detailed physicochemical characterization of these complexes, particularly in the context of LNP-based delivery systems for translational applications.

In the Conclusion section we have included:

(revised manuscript)

In this study, DNA-NLS conjugates were delivered via lipofection, a well-established in vitro transfection method. For clinical translation, a natural progression will be to incorporate DNA-NLS conjugates into lipid nanoparticles (LNPs), which represent the leading non-viral delivery modality for nucleic acid therapeutics. The chemical and structural compatibility of DNA-NLS conjugates with established LNP formulation strategies should be further evaluated to determine if this approach is readily adaptable for in vivo delivery. Future studies will focus on optimizing DNA-NLS LNP formulations and evaluating their efficacy and biodistribution in murine models.

Regarding the claimed “generability” of DNA-peptide conjugates and their “potential for diverse

nucleic acid delivery applications,” the reader may question the effect of alternative gene transfer agents or other delivery methods, such as electroporation.

[Authors reply] We thank the reviewer for this thoughtful comment regarding alternative gene transfer agents, such as electroporation. Using a Neon NxT Electroporation System, we attempted to deliver our DNA–NLS conjugates into growth-arrested HepG2 cells. However, under multiple electroporation conditions recommended for HepG2 cells, we observed extensive cell death (see representative images below). We attribute this to the combined stress of electroporation and growth arrest, which likely impaired cell reattachment and survival, preventing meaningful assessment of gene expression.

Maintaining growth-arrested conditions is essential for accurately evaluating NLS-mediated nuclear translocation. Further, as electroporation can transiently disrupt the nuclear envelope, this method for DNA delivery may complicate interpretation of NLS-specific nuclear translocation (Cervia *et al.*, *Mol. Ther. Nucleic Acids*, **11**, 263-271, 2018). Consequently, we believe lipofection remains the most appropriate *in vitro* delivery method for assessing nuclear import mechanisms for this study. For translational applications, we anticipate that lipid nanoparticle (LNP) based systems will provide a clinically relevant route for non-viral delivery of DNA–NLS conjugates, as discussed in the revised Conclusion section.

(revised manuscript)

In this study, DNA-NLS conjugates were delivered via lipofection, a well-established *in vitro* transfection method. For clinical translation, a natural progression will be to incorporate DNA-NLS conjugates into lipid nanoparticles (LNPs), which represent the leading non-viral delivery modality for nucleic acid therapeutics. The chemical and structural compatibility of DNA-NLS conjugates with established LNP formulation strategies should be further evaluated to determine if this approach is readily adaptable for *in vivo* delivery. Future studies will focus on optimizing DNA-NLS LNP formulations and evaluating their efficacy and biodistribution in murine models.

Toxicity is mentioned some times in the paper, on pages 2 and 12. The authors emphasize the importance to reduce the quantity of transgene to limit any possible side-effect. On page 23, it can be read “A concentration of 10 μM was identified as optimal, effectively inhibiting importin- β while maintaining cell viability” and in Supplementary Information, it is reported “Among the tested concentrations, 10 μM importazole exhibited the strongest importin- β inhibition while maintaining cell viability. Higher concentrations (20 and 25 μM) resulted in increased cytotoxicity.”. However, no supporting data related to cell viability or cytotoxicity is provided.

[Authors reply] We thank the reviewer for this comment regarding cell viability following importazole treatment. As shown in Supplementary Fig. 14, we provide representative images of cell nuclei after importazole exposure. Nuclear morphology (size and integrity) was used as a proxy for cell viability, as nonviable cells exhibit distinct nuclear condensation and fragmentation. Further, the importazole concentrations used in our experiments are consistent with those previously reported to maintain cell viability (Soderholm *et al.*, *ACS Chem. Biol.* **6**, 700–708, 2011).

While the authors evaluated the knockout of SV40, I wonder why a KO of PLSCR-1 was not investigated, given that the latter yielded the best results across most conditions.

[Authors reply] We thank the reviewer for this comment regarding a potential knockout of the PLSCR-1 NLS peptide. To our knowledge, no defined knockout of mutant variant of the PLSCR-1 NLS has been reported. The SV40 NLS KO (K \rightarrow T mutation) is well-characterized and validated

in numerous studies for its ability to completely inhibit nuclear translocation through the nuclear pore complex, serving as an effective negative control for our study.

Some sentences should be rephrased for more clarity. For instance: “cytosolic and secreted DNA payloads” in the abstract sounds confusing.

[Authors reply] We thank the reviewer for this comment. We have rephrased sentences and statements throughout the manuscript to improve clarity.

What are Q3 and Q4 in Supplementary Figures 4 and 6?

[Authors reply] We thank the reviewer for this comment. Q3 and Q4 are artifacts from the quadrant tool used during flow cytometry analysis to set the thresholding to define eGFP positive vs eGFP cell populations. For clarity, these quadrant labels have been removed from Supplementary Figures 4 and 6 (now 11 and 13).

Reviewer #3 (Remarks to the Author):

Delivery of DNA to the nucleus of non-proliferating cells is critical for gene expression but has been challenging in gene therapy. This manuscript reported a new strategy to increase the efficiency by generating DNA oligonucleotide-peptide conjugates. Using a GFP reporter, they optimized the design and screened multiple NLS peptides in growth-arrested human cell lines. They also evaluated the generality of the approach using other gene cassettes. This study provides a new strategy for non-viral gene therapy. However, more evidence is required to support the findings.

[Authors reply] We appreciate the positive comments from the reviewer. We have carefully considered and addressed the concerns as detailed below.

1. The major concern is that the entire study relies solely on measuring protein product levels. Direct evidence is needed to quantify the distribution of DNA between the nucleus and cytoplasm. Additionally, the total level of transfected DNA in cells has not been quantified, making it important to rule out the possibility that different constructs affect transfection efficiency. Finally, RNA levels should be measured to confirm that enhanced nuclear import of DNA correlates with increased transcription.

[Authors reply] We thank the reviewer for this insightful comment emphasizing the need for direct evidence of increased nuclear translocation of NLS-modified gene cassettes, transfection efficiency, and mRNA transcripts levels. To address this, we transfected NLS-modified gene cassettes and relevant controls (plasmid, linear control, and SV40KO-modified cassette) into growth-arrested HepG2 cells. 48 hours post lipofection, cells were gently lysed, and nuclear and extranuclear fractions were isolated. Quantitative PCR was then performed to determine the copy number of transfected DNA in each fraction. We observed a significant ($p < 0.05$) increase (~10–20 fold) in nuclear DNA copies in cells transfected with NLS-modified cassettes compared to controls. This provides strong evidence that NLS modification enhances nuclear import of DNA cargo, consistent with the elevated expression levels previously reported.

Additionally, total transfected DNA and mRNA transcripts were quantified by qPCR across different gene cassettes, further supporting that NLS conjugation improves nuclear translocation and subsequent transcription. We observed no differences in the amount of DNA delivered to cells between peptide-modified and unmodified gene cassettes. As anticipated, mRNA levels corresponded with observed eGFP expression, with NLS-modified gene cassettes resulting in increased RNA transcripts.

Based on these findings, we have revised Figure 3 to include the nuclear DNA copy number data (now presented as Fig. 3d) and expanded the discussion accordingly.

Figure 3. Optimization and Nuclear Localization of NLS-modified gene cassettes

(a) Parameters of NLS-modified eGFP gene cassettes were optimized: $(GGGGS)_n$ linker length (L_n), single/dual(MIuI/AvrII) conjugation, and peptide sequences. Linear eGFP cassettes were transfected into aphidicolin growth arrested HepG2 cells, and eGFP expression was monitored by microscopy and flow cytometry. (b) Representative epifluorescence microscopy images showing eGFP expression in growth-arrested HepG2 cells 48 hours after transfection with linearized eGFP

cassette. Shown are capped, linearized cassettes with no peptide (linear control), an SV40 KO peptide (a single amino acid substitution, K→T, from the SV40 NLS that is known to inhibit nuclear translocation), and three example NLS peptides. “Linear control” is the capped, unmodified linear eGFP cassette. Scale bar: 50 μm. (c) Percent of growth-arrested HepG2 cells with eGFP fluorescence 48 hours after transfection with NLS-modified DNA cassettes in which the NLS (SV40 or Influenza) is on the 5′ MluI side (peptide A), 3′ AvrII side (peptide B), or both sides (peptides A and B). The “SV40 KO” negative control is the linear cassette modified on the 5′ MluI side with the SV40 KO peptide. All peptides utilize the 3x linker ((GGGGS)₃). (d) Copy number of the eGFP DNA gene cassette in the nucleus (per cell) 48 hours after lipofection. Peptide modifications are on the 5′ MluI side of the gene cassette with the 3x linker ((GGGGS)₃). Controls include eGFP plasmid, unmodified, capped linear cassette, and the SV40 KO modified cassette. (e) Percent of growth-arrested HepG2 cells with eGFP fluorescence 24 hours after transfection with eGFP cassettes with or without 10 μM importazole treatment. Peptide modifications are on the 5′ MluI side of the gene cassette with the 3x linker ((GGGGS)₃). Controls include eGFP plasmid, unmodified, capped linear cassette, and the SV40 KO modified cassette. Data collected in (c) and (e) were quantified using flow cytometry and are presented as mean ± s.d. for $n=5$ biologically independent experiments, with individual data points shown, 100 ng DNA transfected per condition. Data collected in (d) were quantified using qPCR, normalized against RPP30 housekeeping gene, and are presented as mean ± s.d. for $n=3$ biologically independent experiments, with individual data points shown, 100 ng DNA transfected per condition. For (c) and (d), statistical analysis was performed using one-way ANOVA with Tukey’s multiple comparison ($*p \leq 0.05$, $**p \leq 0.01$, $***p \leq 0.001$, $****p \leq 0.0001$, $ns p > 0.05$). For (e), statistical analysis was performed using multiple unpaired t -tests ($*p \leq 0.05$, $**p \leq 0.01$, $***p \leq 0.001$, $****p \leq 0.0001$, $ns p > 0.05$).

(revised manuscript)

We quantified the copy number of exogenous DNA translocated to the nucleus and found that NLS-modified gene cassettes had significantly enhanced nuclear delivery compared to controls.

While these results strongly suggest that NLS-modified DNA cassettes enhance expression by promoting nuclear translocation, they do not directly demonstrate increased nuclear import. To quantify total DNA delivery to growth-arrested HepG2 cells, we lysed cells 48 hours post-transfection and performed quantitative PCR (qPCR) to determine absolute DNA copy numbers. There was no significant difference ($p > 0.05$) in total intracellular DNA between NLS-modified and unmodified cassettes, indicating that enhanced gene expression from NLS-modified constructs is not simply due to increased cellular uptake via lipofection (Supplementary Figs. 7, 8). However, qPCR analysis of isolated nuclear fractions revealed a significant ($p < 0.05$) enrichment (~10-20 fold) of NLS-modified DNA compared to controls (Fig. 3d), providing direct evidence that NLS conjugation facilitates nuclear import of DNA cargo. The observed increase in nuclear copy number, from hundreds to thousands of DNA molecules per nucleus, aligns with previously reported thresholds required for robust gene expression.^{20,57–60} This correlation is in agreement with the elevated gene expression we observed for NLS-modified gene cassettes. Consistent with this, eGFP mRNA quantification from transfected cells confirmed that enhanced nuclear accumulation of NLS-modified DNA cassettes corresponds to both higher transcription and subsequent expression (Supplementary Fig. 9).

Additionally, the following Supplementary Figures regarding total transfected DNA levels and mRNA transcription were included:

Supplementary Figure 7. Total eGFP DNA cassette copy number delivered to growth-arrested HepG2 cells 48 h following lipofection.

Data collected were quantified using qPCR, normalized against RPP30 as a housekeeping gene, and are presented as mean \pm s.d. for $n = 5$ biologically independent experiments, with individual data points shown, 100 ng DNA transfected per condition. Statistical analysis was performed using one-way ANOVA with Tukey's multiple comparison ($*p \leq 0.05$, $**p \leq 0.01$, $***p \leq 0.001$, $****p \leq 0.0001$, $ns p > 0.05$).

Supplementary Figure 8. Extranuclear, nuclear, and total eGFP DNA gene cassette copy numbers delivered to growth-arrested HepG2 cells 48 h following lipofection. The nuclear fraction of lysate was separated from the extranuclear fraction via centrifugation (see methods).

Data collected were quantified using qPCR, normalized against RPP30 as a housekeeping gene, and are presented as mean \pm s.d. for $n = 5$ biologically independent experiments, with individual data points shown, 100 ng DNA transfected per condition. Nuclear DNA copies are also presented in Figure 3d. Total delivered DNA copies is a summation of extranuclear and nuclear DNA copies. Statistical analysis was performed using one-way ANOVA with Tukey's multiple comparison ($*p \leq 0.05$, $**p \leq 0.01$, $***p \leq 0.001$, $****p \leq 0.0001$, $ns p > 0.05$).

Supplementary Figure 9. mRNA transcripts copy numbers from eGFP DNA gene cassettes delivered to growth-arrested HepG2s via lipofection.

Data collected were quantified using qPCR, normalized against GAPDH as a housekeeping gene, and are presented as mean \pm s.d. for $n = 5$ biologically independent experiments, with individual data points shown, 100 ng DNA transfected per condition. Statistical analysis was performed using one-way ANOVA with Tukey's multiple comparison ($*p \leq 0.05$, $**p \leq 0.01$, $***p \leq 0.001$, $****p \leq 0.0001$, $ns p > 0.05$).

We believe the inclusion of these experiments and the data generated provides strong evidence supporting that our NLS-modifications on gene cassettes facilitate greater nuclear import of a DNA cargo.

2. The description and diagram of the method are not clear. How is the addition of single versus dual peptides to the gene cassettes controlled?

[Authors reply] We thank the reviewer for this comment. The addition of single versus dual peptides to the gene cassettes is controlled through the use of two distinct restriction enzyme sites located at opposite ends of the PCR-amplified gene cassette. Specifically, digestion with MluI and AvrII generates unique 5' and 3' sticky ends, respectively. DNA stem loops were designed with complementary overhangs corresponding to each restriction site and subsequently ligated to the linearized cassette to generate a capped construct. Each DNA stem loop can be either unmodified or peptide-modified prior to ligation, enabling precise control over the placement of peptide

modifications. By selectively incorporating a peptide-modified stem loop at one or both ends, we can generate single or dual peptide-modified gene cassettes in a site-specific manner.

To further clarify the strategy to generate peptide-modified gene cassettes we have made the following changes.

(original manuscript)

To generate peptide-modified gene cassettes, we chose to utilize a capped linear gene cassette, similar to previous work, that can be modified with peptides to evaluate nuclear uptake of our DNA-NLS constructs (Figure 2d).²⁷ The eGFP gene cassette flanked by two unique restriction sites, AvrII and MluI, was generated via PCR amplification and subsequent digestion, yielding a linear eGFP fragment with defined sticky ends. Our TGT-recognizable DNA oligonucleotide stem loops were designed such that they contain 5' overhangs complementary to these sticky ends, enabling ligation to the digested eGFP fragment to generate a capped eGFP gene cassette which was verified by agarose gel electrophoresis (Supplementary Fig. 3). Using the workflow described, we can modify either or both DNA oligonucleotide stem loops (A and/or B) with azido-peptides, effectively generating single- or dual-peptide-modified eGFP gene cassettes.

(revised manuscript)

To generate peptide-modified gene cassettes, we chose to utilize a capped linear gene cassette, similar to previous work, that can be selectively modified with peptides to evaluate nuclear uptake of our DNA-NLS constructs (Figure 2d).²⁷ A eGFP gene cassette flanked by two unique restriction sites, AvrII and MluI, was generated via PCR amplification and subsequent digestion, yielding a linear eGFP fragment with defined sticky ends by the restriction enzymes used. Our TGT-recognizable DNA oligonucleotide stem loops were designed such that they contain 5' overhangs complementary to these sticky ends (either MluI or AvrII), enabling ligation to the digested eGFP fragment to generate a capped eGFP gene cassette which was verified by agarose gel electrophoresis (Supplementary Fig. 3). Using the workflow described, we can modify either or both DNA oligonucleotide stem loops (A and/or B) with azido-peptides prior to ligation, effectively generating single- or dual-peptide-modified eGFP gene cassettes following ligation.

3. It is not clear whether the linear control DNA contains the end loops. Again it is best to have a diagram of the individual construct used in the paper. And Figure 1 (especially 1a) is more suitable for supplemental information.

[Authors reply] We thank the reviewer for this comment regarding the clarity of the linear control constructs. The linear control represents a peptide-free, capped linear cassette in which the same DNA stem loops are ligated to the sticky-ended gene cassette; however, these stem loops are unmodified and do not contain peptides. Several clarifications have been added throughout the manuscript to make this explicit.

(revised manuscript)

Negative control samples (capped linear gene cassettes containing no peptide) were generated by ligating unmodified DNA stem loops to both ends of the sticky-ended linear DNA fragment.

We observed higher eGFP expression from DNA constructs modified with NLS peptides (Fig. 3b) compared to the linear control, which consisted of the same gene cassette ligated to DNA stem loops lacking peptide modification.

“Linear control” is the capped, unmodified linear eGFP cassette.

We believe that Fig. 1a is appropriately placed in the main manuscript, as it provides essential context by summarizing previous strategies for generating DNA–NLS conjugates and illustrating potential reasons for their inconsistent success in enhancing nuclear translocation. Additionally, Fig. 1b serves as a concise schematic of our DNA-NLS gene cassette, emphasizing the key design distinctions that contribute to its improved performance in facilitating nuclear import.

4. It is unclear how this approach could be applied clinically—does it require transfection of DNA–peptide conjugates?

[Authors reply] We thank the reviewer for this insightful comment. We do not anticipate lipofection to be a suitable method for (pre)clinical delivery of DNA-NLS conjugates. Lipofection was employed in this study due to its well-established reliability for in vitro transfection and mechanistic evaluation of DNA delivery. We view the present work as a proof-of-principle study demonstrating the versatility of the DNA-PepTAG platform and showing that NLS-modified gene cassettes exhibit increased expression through enhanced nuclear translocation.

Looking ahead, we are particularly excited about adapting this approach for in vivo applications using lipid nanoparticle (LNP) formulations, which represent the current standard for non-viral nucleic acid delivery. The modular chemistry of DNA-PepTAG makes it readily compatible with established LNP formulation strategies, offering a promising path toward translating DNA-NLS conjugates into therapeutic contexts. Future studies will focus on optimizing DNA-NLS LNP formulations and evaluating their efficacy and safety in preclinical models. To better reflect these ideas we have changed the discussion in the revised manuscript:

(original manuscript)

Beyond NLS modification, this platform is broadly adaptable, allowing for the conjugation of DNA with other peptides or azide-containing moieties, expanding its potential for diverse nucleic acid delivery applications. Future studies will focus on optimizing DNA-PepTAG for in vivo applications using clinically relevant delivery modalities such as lipid nanoparticles (LNPs) and assessing delivery efficacy in murine models. Moreover, expanding the screening of NLS peptides from diverse sources may allow for more refined nuclear-targeting strategies, particularly for tissue- or cell-type-specific delivery. The ability to rationally or synthetically design novel peptides tailored for DNA transport further enhances the potential of this approach.

(revised manuscript)

In this study, DNA-NLS conjugates were delivered via lipofection, a well-established in vitro transfection method. For clinical translation, a natural progression will be to incorporate DNA-NLS conjugates into lipid nanoparticles (LNPs), which represent the leading non-viral delivery modality for nucleic acid therapeutics. The chemical and structural compatibility of DNA-NLS conjugates with established LNP formulation strategies should be further evaluated to determine if this approach is readily adaptable for in vivo delivery. Future studies will focus on optimizing DNA-NLS LNP formulations and evaluating their efficacy and biodistribution in murine models. Moreover, expanding the screening of NLS peptides from diverse sources may allow for more refined nuclear-targeting strategies, particularly for tissue- or cell-type-specific delivery. The ability to rationally or synthetically design novel peptides tailored for DNA transport further enhances the potential of this approach. Beyond NLS modification, this platform is broadly adaptable, allowing for the conjugation of DNA with other peptides or azide-containing moieties, expanding its potential for diverse nucleic acid delivery applications.

REVIEWERS' COMMENTS

Reviewer #1 (Remarks to the Author):

I am satisfied with the changes made by the authors, and congratulate them on their work.

[Authors reply] We thank the reviewer for their time in reviewing our manuscript and appreciate their feedback in strengthening the work.

Reviewer #2 (Remarks to the Author):

MINOR REVISION RECOMMENDED

In their revised manuscript entitled "Highly Efficient Expression of DNA-peptide Conjugates in Growth-arrested Cells", Zulfiqar Y. Mohamedshah et al. have addressed most of my comments:

Importantly, the authors conducted comprehensive complementary experiments using quantitative PCR to provide more direct evidence of active nuclear transport. The findings demonstrate that NLS-modified gene cassettes indeed exhibit significantly enhanced nuclear copy number relative to control constructs.

As also requested, eGFP expression from NLS-modified gene cassettes in both nocodazole-arrested and actively dividing HepG2 cells was assessed. The results show that specific growth arrest agent does not affect nuclear translocation of DNA cargo and growth arrest is essential for isolating and assessing NLS-mediated nuclear import.

In addition, DNA-NLS conjugates were compared with those formed through nonspecific electrostatic interactions. The results clearly demonstrate that enhanced gene expression via nuclear translocation requires a single, site-specific, and stable covalent conjugation between DNA and the NLS peptide, rather than noncovalent or labile attachment strategies.

My comment regarding FACS analysis was considered through the use of transfected plasmids, capped linear gene cassettes, and peptide-modified gene cassettes that do not encode fluorescent proteins. These controls confirm that the gating strategy and experimental design are not biased by artifacts. That said, I would have appreciated the inclusion of double fluorescence gating.

The generalizability of the approach has been appropriately discussed. A more consolidated message involving primary or stem cells would have further strengthened the manuscript, although the authors provide a reasonable rationale for their current focus.

The remaining comments I previously raised have also been addressed.

Overall, the manuscript has been substantially improved, with the inclusion of additional Figures or supplementary data and an expanded discussion that reinforces the claims presented in the title.

At this stage, I have only minor comments remaining. These include clarifications such as

specifying “nuclear copy number per cell” on the Y-axis of Figure 3d, and enhancing Figure quality and font size to improve readability.

[Authors reply] We have updated the axis labeling for Figure 3d and enhanced the Figure quality for readability.

Accordingly, I consider the manuscript suitable for publication in Nature Communications, pending minor revisions to address these final points.

[Authors reply] We thank the reviewer for their time in reviewing our manuscript and appreciate their feedback in strengthening the work.

Reviewer #3 (Remarks to the Author):

The manuscript is significantly improved with additional supporting evidence. Measurements of DNA copies in total and nuclear fractions, along with RNA transcript quantification, strengthen the validity of this new approach.

[Authors reply] We thank the reviewer for their time in reviewing our manuscript and appreciate their feedback in strengthening the work.

Minor editing issues:

In the text “the dual-end modification drastically reduced expression to levels below that of the SV40 KO control (<1%) (Figure 3d, Supplementary Fig. 5)”, it is not appropriate to cite 3d here?

[Authors reply] We have now properly referenced the appropriate SI Figure only.

In Fig. 3d: it is better to directly label “Nuclear DNA copy number per cell” on the panel to avoid confusion.

[Authors reply] We have updated the axis labeling for Figure 3d.